# Photochemical environment over Southeast Asia primed for hazardous ozone levels with influx of nitrogen oxides from seasonal biomass burning

Margaret R. Marvin[1], Paul I. Palmer[1,2], Barry G. Latter[3,4], Richard Siddans[3,4], Brian J. Kerridge[3,4], Mohd Talib Latif[5], and Md Firoz Khan[6]

[1]National Centre for Earth Observation, University of Edinburgh, Edinburgh, UK
[2]School of GeoSciences, University of Edinburgh, Edinburgh, UK
[3]Remote Sensing Group, STFC Rutherford Appleton Laboratory, Chilton, UK
[4]National Centre for Earth Observation, STFC Rutherford Appleton Laboratory, Chilton, UK
[5]Department of Earth Sciences and Environment, Faculty of Science and Technology, Universiti Kebangsaan Malaysia, Bangi, Malaysia
[6]Department of Chemistry, Faculty of Science, Universiti Malaya, Kuala Lumpur, Malaysia

**Correspondence:** Margaret R. Marvin (mmarvin@ed.ac.uk)

**Abstract.** Mainland and maritime Southeast Asia are home to more than 655 million people, representing nearly 10% of the global population. The dry season in this region is typically associated with intense biomass burning activity, which leads to a significant increase in surface air pollutants that are harmful to human health, including ozone ($O_3$). Latitude-based differences in dry season and land use distinguish two regional biomass burning regimes: (1) burning on the peninsular mainland peaking in March and (2) burning across Indonesia peaking in September. The type and amount of material burned in each regime impacts the emissions of nitrogen oxides ($NO_x$ = NO + $NO_2$) and volatile organic compounds (VOCs), which combine to produce ozone. Here, we use the nested GEOS-Chem atmospheric chemistry transport model (horizontal resolution of $0.25° \times 0.3125°$), in combination with satellite observations from the Ozone Monitoring Instrument (OMI) and ground-based observations from Malaysia, to investigate ozone photochemistry over Southeast Asia in 2014. Seasonal cycles of tropospheric ozone columns from OMI and GEOS-Chem peak with biomass burning emissions. Compared to OMI, the model has a mean annual bias of −11% but tends to overestimate tropospheric ozone near areas of seasonal fire activity. We find that outside of these burning areas, the underlying photochemical environment is generally $NO_x$-limited, dominated by anthropogenic $NO_x$ and biogenic non-methane VOC emissions. Pyrogenic emissions of $NO_x$ play a key role in photochemistry, shifting towards more VOC-limited ozone production and contributing about 30% of the regional ozone formation potential during both biomass burning seasons. Using the GEOS-Chem model, we find that biomass burning activity coincides with widespread ozone exposure at levels that exceed world public health guidelines, resulting in about 260 premature deaths across Southeast Asia in March of 2014 and another 160 deaths in September. Despite a positive model bias, hazardous ozone levels are confirmed by surface observations during both burning seasons.

# 1 Introduction

Mainland and maritime Southeast Asia, including Myanmar, Laos, Vietnam, Cambodia, Thailand, Malaysia, Indonesia, Singapore, Brunei, Timor-Leste and the Philippines, is home to more than 655 million people. Its inhabitants are mainly distributed in densely packed cities with sizes that range from <1 million people to megacities of more than 10 million people. The geographical region is a tightly packed mosaic of arable land, forest and cities that all emit large amounts of trace gases with e-folding lifetimes that allow them to mix and be transported across the region. Urban populations are growing at a rate faster than global mean values, and future projections suggest this trend will continue, with massive implications for regional anthropogenic emissions. These changes in urbanisation occur against the backdrop of widespread seasonal biomass burning and significant biogenic emissions. Here, we use a nested regional 3-D model of atmospheric chemistry to explore the competing roles of ozone precursors from different source sectors and how they collectively determine observed variations of tropospheric ozone in Southeast Asia, particularly during regional fire seasons.

Nearly one half of the populated area across Southeast Asia is exposed to recurring fire activity every year (Vadrevu et al., 2019), with implications for surface air pollution and consequently human health. Over this region, fires originate predominately from land use change (Marlier et al., 2015) and seasonal agricultural practices (Sastry, 2002; Jones, 2006; Korontzi et al., 2006; Herawati and Santoso, 2011). Subannual variability in fire activity peaks during the dry season, which spans November–May on mainland Southeast Asia but is shifted into June–October for Indonesia and other countries near the equator (Duncan et al., 2003; Csiszar et al., 2005). Fire activity in this region is also subject to significant interannual variability due to large-scale climate variations, e.g. El Niño Southern Oscillation, which have culminated in some of the most severe fire events in Southeast Asia on record (van der Werf et al., 2008; Wooster et al., 2012; Marlier et al., 2013), including the extreme Indonesian fires of 2015 (Field et al., 2016; Huijnen et al., 2016).

There are a number of air pollutants that contribute to unhealthy air quality over Southeast Asia. Here, we focus on tropospheric ozone ($O_3$). Human exposure to ozone causes cardiovascular and respiratory illness, and it has been linked with increased mortality (Samet et al., 2000; Bell et al., 2004; Vicedo-Cabrera et al., 2020). Ozone is produced through the secondary photochemistry of $NO_x$ ($NO_x = NO + NO_2$) and volatile organic compounds (VOCs) (Trainer et al., 1987; Jacob, 1999; Thornton et al., 2002). These ozone precursors are emitted directly from anthropogenic, biogenic, and pyrogenic sources (Stettler et al., 2011; Guenther et al., 2012; Li et al., 2017; Hoesly et al., 2018). The photochemistry that links $NO_x$ and VOC emissions to ozone is complex and non-linear (Sillman et al., 1990; Sillman, 1999), which makes it particularly challenging to interpret the mechanisms that relate regional fire activity to surface ozone air quality.

Previous studies have established a broad connection between biomass burning and enhanced ozone over Southeast Asia (Kita et al., 2000; Pochanart et al., 2001; Thompson et al., 2001; Ziemke et al., 2009; Toh et al., 2013). Due to a scarcity of ground-based measurements, many of these studies utilize satellite observations of tropospheric ozone, which provide greater spatial and temporal coverage than surface *in situ* measurements but do not necessarily represent the photochemical sensitivity of surface ozone to precursor emissions. Others have applied atmospheric chemistry models to simulate the impact of biomass

burning on ozone air quality throughout the region, but evaluation of existing model studies against ozone observation datasets is limited.

We use a nested version of the GEOS-Chem model of atmospheric chemistry and transport to investigate ozone photochemistry over mainland and maritime Southeast Asia. We focus our analysis on 2014 because it represents a year in which there was moderate burning, occurring before the 2015/2016 El Niño that significantly impacted the region. We evaluate our model using satellite observations of tropospheric ozone from the Ozone Monitoring Instrument (OMI) and ground-based observations of surface ozone from Malaysia. In the next section, we describe these data and the GEOS-Chem model. In Sect. 3, we describe the seasonal distributions of biomass burning, $NO_x$ and VOC emissions from pyrogenic, anthropogenic and biogenic activity, and the resulting distributions of tropospheric ozone. In Sect. 4, we use a variety of indicators to quantify the sensitivity of tropospheric ozone to changes in biomass burning emissions. We use the maximum daily 8-hour average (MDA8) metric in Sect. 5 to assess the impact of ozone on public health. We conclude the paper in Sect. 6.

## 2 Data and methods

Here, we describe satellite and ground-based observations of tropospheric ozone from mainland and maritime Southeast Asia in 2014, and we describe the GEOS-Chem model of atmospheric chemistry and transport that we use to interpret these data.

### 2.1 Tropospheric ozone columns from the Ozone Monitoring Instrument

For our analysis we use spaceborne measurements of ozone from the Ozone Monitoring Instrument (OMI) (Levelt et al., 2006, 2018), launched in 2004 aboard the NASA Aura satellite, with a sun-synchronous orbit and a local equator crossing time of 1345 ($\pm15$ mins). OMI uses two imaging grating spectrometers each with a CCD detector to collect solar backscattered radiation in the spectral range 270–500 nm using three channels: UV-1 (264–311 nm), UV-2 (307–383 nm) and Vis (349–504 nm). OMI has an across-track swath width of 2600 km, which in global mode is described by 60 scenes that have ground footprints from $13\times24$ km$^2$ at nadir to $28\times160$ km$^2$ at the swath edges.

We use ozone profiles retrieved by the RAL Remote Sensing Group using a multi-step scheme (fv0214) based on that developed for GOME-class sensors (Miles et al., 2015). The fv0214 scheme uses optimal estimation to retrieve ozone on a fixed set of pressure levels (hPa): surface pressure, 450, 170, 100, 50, 30, 20, 10, 5, 3, 2, 1, 0.5, 0.3, 0.17, 0.1, 0.05, 0.03, 0.017 and 0.01. Subcolumns are calculated for the layers between levels. In the first step, the sun-normalised radiance spectrum in the Hartley band (270–306 nm), binned at $50\times50$ km$^2$ to achieve adequate photometric signal to noise, is fitted to provide information principally on the stratospheric profile. Information on tropospheric ozone is obtained in a subsequent step by fitting temperature-dependent spectral structure in the Huggins bands (325–335 nm) to < 0.1% RMS precision. Due to computational constraints, only one of the four ground pixels in each $50\times50$ km$^2$ bin is processed in the Huggins bands step, and associated diagnostics such as the *a posteriori* error covariance and averaging kernel matrices are stored for only one in four of the $50\times50$ km$^2$ bins. Beyond basic quality control against retrieval convergence criteria, additional screening is carried out to remove retrieved ozone profiles affected by a wide viewing angle (line-of-sight zenith angle < 55°) or cloudy conditions

(product of cloud fraction and cloud top height < 1 km). Profiles retrieved from rows 17 and 23 of the 2-D detector array are also excluded. This filtering procedure retains 30–40% of pixels retrieved during the months of peak biomass burning in Southeast Asia in 2014.

Reported retrieval errors are standard deviations estimated as the square root of the diagonal elements of the *a posteriori* error covariance matrix with contributions from the radiance measurement and the *a priori* (Miles et al., 2015). The vertical sensitivity of the retrieval at each layer is provided by the averaging kernel matrix, which specifies the perturbation to retrieved ozone at the given layer with respect to that in the true ozone profile on a set of more finely spaced layers used by the radiative transfer model. In this study, we use data for the surface–450 hPa layer, which is well-resolved from the stratosphere. Figure 1 shows an example good-quality OMI ozone profile up to 5 hPa retrieved over mainland Southeast Asia (lat = 15.45° N, lon = 103.91° E) at the local satellite overpass time (solar zenith angle = 28.79°) on 4 March 2014. Figure 1a shows that the tropospheric ozone column retrieved in this example profile is about $5 \times 10^{17}$ molec cm$^{-2}$, and the estimated retrieval error is 32% of that value. Figure 1b shows that the averaging kernel for the surface–450 hPa layer of the same profile is greater than 0.7 in that vertical range, with increasing sensitivity towards the daytime planetary boundary layer (PBL).

## 2.2 Ground-based observations of surface ozone in Malaysia

We use measurements of surface ozone from the Air Quality Division, Department of Environment, Ministry of Environment and Water of Malaysia. These measurements are collected as part of a country-wide air quality monitoring network that has been in place since 1997. Monitoring stations are located 2.5 m above the ground and measure hourly concentrations of gas and aerosol species that are important to air quality, as described by Latif et al. (2014). Ozone is measured by the Teledyne API Model 400/400E (http://www.teledyne-api.com), which detects ozone concentrations by UV absorption at 254 nm. This method has a precision of 0.5% and a detection limit of 0.4 ppb. We use data from all 41 ground stations in this network (Sect. 5) that collected measurements of surface ozone in 2014.

## 2.3 The GEOS-Chem model of atmospheric chemistry and transport

We run version 12.5.0 of the nested 3-D GEOS-Chem atmospheric chemistry transport model (www.geos-chem.org) to describe atmospheric composition over Southeast Asia in 2014. We configure the nested model to provide hourly output at a horizontal resolution of 0.25° × 0.3125° across a domain spanning −10 to 24° N and 90 to 140° E (Fig. 2b–e). We run the nested model at high resolution only for select months from 2014 that demonstrate seasonal patterns in regional biomass burning activity, as described in Sect. 3.1.

The nested model is driven by assimilated meteorology from the GEOS Forward Processing (GEOS-FP) product from the Global Modeling and Assimilation Office (GMAO) at NASA Goddard Space Flight Center. The native resolution of these analyses are 0.25° × 0.3125°, extending through 72 vertical terrain-following sigma-levels that describe the atmosphere from the surface to 0.01 hPa. In GEOS-Chem, the vertical grid is condensed into 47 levels, of which about 30 are typically below the dynamic tropopause.

Initial conditions and time-dependent lateral boundary conditions for the nested model are taken from a self-consistent, global version of the model, run with a coarser spatial resolution of $2° \times 2.5°$ from January 2013 through December 2014. The first year of this global run comprises the model spin up period, which is driven by meteorology from the Modern-Era Retrospective analysis for Research and Applications version 2 (MERRA2), provided by the GMAO. For 2014 we use GEOS-FP meteorology, consistent with the nested model simulation.

For all the simulations we report in this study, we represent atmospheric photochemistry using the "complexSOA_SVPOA" GEOS-Chem mechanism. This mechanism includes the full-chemistry "tropchem" mechanism that describes all gas-phase reactions (Eastham et al., 2014) and also represents the photochemical production of SOA and semi-volatile primary organic aerosols (SVPOA) using a combination of explicit aqueous uptake mechanisms (Marais et al., 2016) with a standard volatility basis set scheme (Pye et al., 2010). As in the "tropchem" mechanism, oxidation chemistry is up-to-date with the literature for many VOC species, including isoprene ($C_5H_8$) and monoterpenes ($C_{10}H_{16}$) (Xie et al., 2013; Bates et al., 2014; Lee et al., 2014; Müller et al., 2014; Fisher et al., 2016; Travis et al., 2016; Chan Miller et al., 2017; Marais et al., 2016). A recent version of this mechanism was shown to simulate the VOC oxidation product formaldehyde (HCHO) within 17% of observed mixing ratios in the summertime Southeast United States where isoprene is the dominant non-methane hydrocarbon (Marvin et al., 2017).

Our simulations use global anthropogenic emissions from the Community Emissions Data System (CEDS) (Hoesly et al., 2018) that we substitute with regional emissions for Asia from the MIX inventory (Li et al., 2017). We use ship emissions from CEDS and aircraft emissions from the Aviation Emissions Inventory Code (AEIC) v2.1 (Stettler et al., 2011). Biogenic VOC emissions are calculated online using the Model of Emissions of Gases and Aerosols from Nature (MEGAN) version 2.1 (Guenther et al., 2012). Recent studies suggest that MEGAN overestimates annual mean isoprene emissions in Southeast Asia by about a factor of 2, peaking at a factor of 4 in the tropical rainforests of maritime Southeast Asia during the wet season (Langford et al., 2010; Stavrakou et al., 2014). Emissions of $NO_x$ from soil and lightning are parameterized in GEOS-Chem (Hudman et al., 2012; Murray et al., 2012).

We use pyrogenic emissions from the Global Fire Emissions Database version 4.1 that includes small fires (GFED4.1s, http://globalfiredata.org). The GFED inventory provides monthly dry matter emissions based on satellite observations of fire activity and vegetation coverage from MODIS (Van Der Werf et al., 2017). These dry matter emissions are classified into six different fire types defined by the predominant vegetation burned: (1) savanna, (2) boreal forest, (3) temperate forest, (4) tropical deforestation, (5) peat and (6) agricultural waste. The GEOS-Chem model applies vegetation-specific emission factors from Akagi et al. (2011) to the dry matter emissions from GFED to produce speciated biomass burning emissions of trace gases and aerosols. Our global simulation utilizes the standard monthly dry matter emissions, but we configure GEOS-Chem to apply daily and three-hourly scaling factors from GFED so that we can resolve finer temporal variability in our nested simulation. For Southeast Asia in 2014, GFED4.1s estimates annual dry matter emissions of 627 Tg, with speciated emissions of $NO_x$ and non-methane hydrocarbons totaling about 1.5 Tg each.

For the purposes of comparison with observations, we sample the model at the time and location of the measurements. To compare directly with the satellite observations, we interpolate model ozone profiles to the vertical levels of the retrievals,

and then we convolve the resulting model ozone subcolumns with scene-dependent averaging kernels, taking into account the *a priori* used by the retrieval.

## 3 Results

### 3.1 Seasonal distributions of biomass burning emissions

Figure 2a shows that there are two distinct fire seasons over Southeast Asia in 2014. The earlier season spans November–May and the later season occurs June–October. Domain-wide monthly dry matter emissions from GFED4.1s (Sect. 2.3) are highest in March ($3.07\times10^{-9}$ kg m$^{-2}$ s$^{-1}$) followed by September ($2.56\times10^{-9}$ kg m$^{-2}$ s$^{-1}$). Seasonal minima occur during May ($8.62\times10^{-11}$ kg m$^{-2}$ s$^{-1}$) and December ($8.72\times10^{-11}$ kg m$^{-2}$ s$^{-1}$).

Figure 2a shows that each burning season has a distinct spatial and compositional distribution of monthly dry matter emis-
160 sions associated with specific combustion processes: deforestation (DEFO), peat (PEAT), savanna (SAVA), agricultural waste (AGRI) and temperate forest (TEMF). There are no emissions from boreal forests (BORF) in this region. Collectively, combustion associated with deforestation, peat, and savanna represent more than 90% of total dry matter emissions over Southeast Asia throughout both burning seasons. Deforestation consistently accounts for 40–50% of all dry matter emissions across the model domain. The remaining contribution is primarily attributed to savanna grasslands in the early burning season and peat-
165 lands later in the year. During the early burning season in March, the distribution of dry matter emissions is 47% DEFO, 14% PEAT, 35% SAVA and 4% other. Figure 2b,d shows that 70% of regional dry matter emissions are generated on the mainland, where the highest emissions ($\sim 1\times10^{-8}$ kg m$^{-2}$ s$^{-1}$) are co-located with the DEFO and SAVA vegetation types. The remaining emissions are primarily concentrated in northern Indonesia of which 50% is due to PEAT. During the later burning season in September, dry matter emissions comprise 42% DEFO, 49% PEAT, 8% SAVA and 1% other. Figure 2c,e shows that 94% of
170 all dry matter emissions originate from Indonesia alone, with highest values in the proximity of DEFO and PEAT fires.

The type, amount, and distribution of burned material determines the chemical composition of emissions and subsequently their impacts on regional air quality. We find that between March and September less than 1% of the monthly dry matter emissions overlap spatially, effectively separating regional biomass burning into two distinct regimes: (1) burning on the peninsular mainland peaking in March and (2) burning in Indonesia peaking in September. The unique spatiotemporal and chemical
distribution of emissions from each regime defines its influence on regional ozone photochemistry.

### 3.2 Seasonal NO$_x$ and VOC emissions

Figure 3 shows a timeseries of monthly mean NO$_x$ and non-methane VOC (NMVOC) emissions (kg m$^{-2}$ s$^{-1}$) from Southeast Asia in 2014, sorted by source sector: anthropogenic, biogenic and pyrogenic. Seasonal patterns of these emissions almost exclusively reflect temporal variations in biogenic and pyrogenic sources, with no discernible seasonal variation from anthro-
180 pogenic emissions. Total emissions peak in March and September when pyrogenic emissions are highest.

The seasonal variation of $NO_x$ emissions, peaking in March and September, reflects the seasonal pattern of dry matter burning emissions as described above. During these two months, pyrogenic emissions of $NO_x$ account for 43% and 32% of total $NO_x$ emissions, respectively. Biogenic emissions of $NO_x$, defined here as originating from soil only, also influence seasonal variations of $NO_x$ emissions over Southeast Asia, but to a much lesser extent than burning. Biogenic emissions peak during April, accounting for only 12% of total $NO_x$ emissions, and are lowest in August–January when the biogenic contribution does not exceed $3.54 \times 10^{-13}$ kg m$^{-2}$ s$^{-1}$, or 3% of the total. Anthropogenic activity represents the dominant source of $NO_x$ in Southeast Asia, accounting for more than 50% of total $NO_x$ emissions in any given month. However, monthly anthropogenic emissions deviate no more than 5% from the annual mean of $9.36 \times 10^{-12}$ kg m$^{-2}$ s$^{-1}$ and therefore have little impact on the seasonal trends of $NO_x$ emissions. Figure 4a–d shows that the spatial distributions of $NO_x$ emissions (kg m$^{-2}$ s$^{-1}$) during March and September are primarily determined by the pyrogenic sector, with a large but approximately constant anthropogenic influence discernible over major cities and ship tracks.

In contrast to $NO_x$ from Southeast Asia, seasonal emissions of NMVOCs are dominated by biogenic activity due mainly to isoprene. Monthly mean NMVOC emissions range from $1.20 \times 10^{-10}$ to $1.82 \times 10^{-10}$ kg m$^{-2}$ s$^{-1}$, with biogenic sources consistently accounting for more than 74% of monthly emissions. Biogenic emissions have a strong seasonal pattern that peaks in May and September during the two regional dry seasons. Pyrogenic NMVOC emissions are comparatively much smaller in magnitude, accounting for 11% and 15% of total NMVOC emissions at their maximum values in March and September, respectively. However, they are still large enough to impact the seasonal trend of total NMVOC emissions, contributing to the domain-wide seasonal maximum in March. Despite lower biomass burning emissions overall, more pyrogenic NMVOC emissions are emitted in September due to the high fraction of organic carbon contained in peat soils (Kumar et al., 2020). Anthropogenic sources contribute only 13% to total monthly mean NMVOC emissions, and do not influence their seasonal variations. Figure 4e–h shows that the spatial distributions of NMVOC emissions (kg m$^{-2}$ s$^{-1}$) in March and September are largely determined by the biogenic sector, the highest emissions originating from rural areas that are predominant with plantation, farming and rainforest canopy. However, total NMVOC emissions may also be enhanced by spatial overlaps with anthropogenic and pyrogenic sources.

## 3.3 Seasonal changes in tropospheric ozone

Figure 5 shows monthly mean GEOS-Chem model and OMI measurements of tropospheric ozone columns (molec cm$^{-2}$) from 2014, averaged across Southeast Asia. Error bars shown for OMI are the standard-errors-in-the-mean, as indication of random error. Also shown are OMI values adjusted for bias estimates with respect to an ozonesonde ensemble, as indication of the systematic error range. [1] Averaging kernels from OMI are applied to profiles from GEOS-Chem in order to produce comparable tropospheric ozone columns (Sect. 2.3).

---

[1]Bias with respect to ozonesondes is derived as a function of latitude and month of year, applying averaging kernels from individual soundings to sonde profiles and averaging over all processed years of the OMI mission. It should be noted, however, that ozonesonde stations are sparse in the latitude range used for this study, particularly in the Southeast Asia longitude sector.

Model tropospheric ozone over Southeast Asia peaks first during March, consistent with precursor emissions of $NO_x$ and NMVOC emissions (Fig. 3), with a value of $4.17 \times 10^{17}$ molec $cm^{-2}$. The second model peak does not occur until October, which we find is due to a large positive flux from atmospheric transport into the region compared with the preceding month (Fig. 6). Minima in model tropospheric ozone occur during August and December, consistent with variations in the precursor emissions. Using the monthly mean tropospheric ozone columns from Fig. 5, we calculate that GEOS-Chem has a mean annual bias of –11% compared to OMI (up to –30% when OMI is adjusted for systematic error). The closest agreement occurs during March when the model has a positive bias of 7% (–27%). Tropospheric ozone from OMI peaks in May with a value of 4.35 (+1.29) $\times 10^{17}$ molec $cm^{-2}$ and not again until October. The annual cycle minimum occurs in January, although there is also a minimum in July between the two seasonal peaks. Broadly speaking, the seasonal trends in tropospheric ozone from GEOS-Chem are consistent with the OMI observations.

Figure 7 shows that tropospheric ozone columns (molec $cm^{-2}$) from both GEOS-Chem and OMI are clearly elevated over areas of seasonal fire activity during March and September of 2014. During March, tropospheric ozone columns increase with latitude, with model values peaking at $8 \times 10^{17}$ molec $cm^{-2}$ over mainland Southeast Asia, where we have shown biomass burning emissions are largest for that time of year (Fig. 2b,d). We find that in March the model has a mean positive bias of 22% over that part of the region. During September, the spatial distribution of tropospheric ozone shifts away from mainland Southeast Asia into Indonesia, again consistent with the seasonal distribution of biomass burning emissions. The model generally has a negative bias with columns rarely exceeding $6 \times 10^{17}$ molec $cm^{-2}$, except for locations where there are emissions from biomass burning (Fig. 2c,e).

## 4 How do biomass burning emissions impact variations in tropospheric ozone?

As shown above, precursors of tropospheric ozone originate from anthropogenic, biogenic and pyrogenic sources. Here, we explore how these emissions influence ozone photochemistry over Southeast Asia using three different ozone sensitivity indicators: 1) ozone production rate, 2) ozone sensitivity to $NO_x$ and VOCs, and 3) ozone formation potential. Figure 8 shows the spatial distributions of all three indicators across Southeast Asia in March and September of 2014. We provide discussion on these indicators in the following sections.

### 4.1 Ozone production rates

Tropospheric ozone is produced when VOCs are oxidized in the presence of $NO_x$. During the daytime, VOCs are oxidized by the hydroxyl radical (OH) to produce peroxy radicals ($HO_2$ and $RO_2$). In the presence of $NO_x$, these peroxy radicals may react with NO to produce $NO_2$, which photolyzes in sunlight to form atomic oxygen. Atomic oxygen and molecular oxygen then rapidly combine to produce ozone. The rate that ozone is produced is limited by the conversion of NO to $NO_2$ and is commonly represented as follows:

$$P(O_3) = k_{HO2+NO}[HO_2][NO] + \Sigma k_{RO2i+NO}[RO_{2i}][NO], \tag{1}$$

where P(O$_3$) is the ozone production rate (molec cm$^{-3}$ s$^{-1}$), $k_{\mathrm{HO2+NO}}$ and $k_{\mathrm{RO2i+NO}}$ are rate constants corresponding to the subscripted reactions (cm$^3$ molecule$^{-1}$ s$^{-1}$), and [HO$_2$], [RO$_{2i}$], and [NO] are species concentrations (molec cm$^{-3}$). The subscript $i$ denotes that the second term is summed over multiple species of organic peroxy radical. We use ozone production rates calculated by GEOS-Chem, which accounts for all contributing species as represented here by the 'complexSOA_SVPOA' mechanism, including RO$_2$ derived from isoprene, monoterpenes and smaller VOCs.

Figure 8a,b shows GEOS-Chem ozone production rates (molec cm$^{-3}$ s$^{-1}$) simulated over Southeast Asia during March and September of 2014. We report the mean daytime value of P(O$_3$) taken across the PBL to account for the influence of local mixing on surface air quality. Based on a qualitative comparison with Fig. 2, the highest ozone production rates, on the order of $2 \times 10^8$ molec cm$^{-3}$ s$^{-1}$, are largely co-located with fire emissions in both March and September. However, ozone production is also elevated over several major cities across the region, where ozone production is dominated by the emission of precursors from the anthropogenic sector. Although the ozone production rate is useful in demonstrating the spatial correlation with biomass burning activity, it provides little insight into the mechanisms responsible for ozone production and the roles of ozone precursors in regional ozone photochemistry.

## 4.2 Ozone sensitivity to NO$_x$ and VOCs

Using GEOS-Chem model output, we calculate tropospheric ozone sensitivity to NO$_x$ and VOCs with the L$_N$/Q indicator, which represents the fraction of radical termination that results in the removal of NO$_x$ relative to HO$_x$ (HO$_x$ = OH + HO$_2$) (Kleinman et al., 1997, 2001). Although both radical families are needed to form ozone, NO$_x$ is far less abundant than HO$_x$, and the loss of NO$_x$ by radical termination effectively inhibits the production of ozone. The major terminal sinks for NO$_x$ and HO$_x$ are nitric acid (HNO$_3$) and hydrogen peroxide (H$_2$O$_2$), respectively. Thus the photochemical production rates for these species (molec cm$^{-3}$ s$^{-1}$) are used to approximate radical termination in the calculation of L$_N$/Q:

$$\mathrm{L}_N/\mathrm{Q} = \frac{\mathrm{P(HNO_3)}}{\mathrm{P(HNO_3)} + \mathrm{P(H_2O_2)}}. \tag{2}$$

Where L$_N$/Q < 0.5, radical termination is mainly controlled by the loss of HO$_x$, and ozone production varies linearly with NO$_x$ in a relationship that is considered NO$_x$-limited. Where L$_N$/Q > 0.5, ozone production is inhibited by the loss of NO$_x$ and becomes more sensitive to VOC emissions in a relationship that is considered VOC-limited. This method of calculating L$_N$/Q assumes a simplified removal scheme for HO$_x$ and NO$_x$ radicals, as both can also be removed through the formation of organic products. Loss of isoprene nitrates to hydrolysis has been shown to provide an important sink for NO$_x$ (Bates and Jacob, 2019). Although the organic sink for HO$_x$ may offset the impact of organic nitrates, formation of organic peroxides is intertwined with complex recycling mechanisms, which makes it difficult to assess the total impact of these uncertainties on our calculation of L$_N$/Q at this time.

Figure 8c,d shows L$_N$/Q calculated from GEOS-Chem model output for March and September of 2014. For consistency with the ozone production rates in Fig. 8a,b, we report mean daytime PBL values of L$_N$/Q. In general, we find that L$_N$/Q increases with NO$_x$ emissions and ozone production rates. The lowest values of L$_N$/Q occur where NO$_x$ emissions are low, even if VOC emissions are very high, such as over forests and croplands where there is no biomass burning activity. In these areas,

$L_N/Q$ is less than 0.5, indicating that ozone production is $NO_x$-limited and therefore more sensitive to emissions of $NO_x$ than to VOC emissions. Values of $L_N/Q$ tend to approach or exceed 0.5 over large cities and areas of biomass burning, indicating that these are instead characteristic of the VOC-limited regime. These areas coincide with the highest ozone production rates in Southeast Asia, suggesting that the majority of regional ozone forms under VOC-limited conditions. Although we cannot quantify sector-specific contributions to ozone production using $L_N/Q$, we can achieve this with our final ozone sensitivity

indicator, the ozone formation potential.

## 4.3   Ozone formation potential

Ozone formation potential (OFP) is an ozone sensitivity indicator that relates ozone production directly to the emission of VOCs. To calculate OFP, we use the maximum incremental reactivity (MIR) scale developed by Carter (1994), where each emitted VOC is assigned a unique MIR that represents both the reactivity of that VOC to oxidation by OH and the capacity

of that VOC once oxidized to produce ozone. The MIR scale is based on yields from the SAPRC photochemical mechanism (Carter, 1990) and has been validated in chamber studies (Carter, 1994, 1999, 2010). Here, we use the most recent MIR values provided by Carter (2010), which are described by the units g ozone formed per g VOC emitted. Because the MIR is a mass-based ratio, we simply multiply the emissions of each VOC (kg m$^{-2}$ s$^{-1}$) by its corresponding MIR to calculate OFP as a positive flux of ozone into the atmosphere (kg m$^{-2}$ s$^{-1}$):

$$OFP_{VOC i} = E_{VOC i} \times MIR_{VOC i}. \tag{3}$$

We acknowledge that the MIR scale is derived by adjusting emissions of $NO_x$ to yield the highest possible incremental VOC reactivity. This means that OFP calculated from MIR assumes ozone production under VOC-limited conditions. Figure 8e,f shows that OFP is generally enhanced over the parts of Southeast Asia that we have identified as VOC-limited. However, we calculate OFP for the entire region, which also includes many areas that may be $NO_x$-limited. We do not filter OFP for

VOC-limited conditions because precursor emissions and $L_N/Q$ vary widely on much finer spatiotemporal scales than are represented here (Bai et al., 2015; Mazzuca et al., 2016; Van Der Werf et al., 2017). Thus the assumption that OFP represents VOC-limited conditions introduces some uncertainties to our analysis, which we address in Sect. 4.4.

Different photochemical properties among source VOCs translate into a wide range of ozone formation capabilities. Table 1 shows VOCs with the highest assigned MIR that are emitted within GEOS-Chem. All of these species contain functional

groups, such as alkenes and aldehydes, that are particularly reactive to oxidation by OH. Many of these are also large molecules containing multiple carbons that can be oxidized further to produce a high yield of ozone. When we apply MIR to NMVOC emission fluxes, we can determine which NMVOC have the largest influence on OFP. Figure 9a shows the source attribution of OFP (kg m$^{-2}$ s$^{-1}$) to NMVOCs over Southeast Asia in March and September of 2014. For both months, we find that the single largest contributor accounting for more than 60% of OFP is isoprene (ISOP in GEOS-Chem), which is emitted in the model

exclusively from biogenic sources. Other major contributors include propene (PRPE), acetaldehyde (ALD2), monoterpenes (MTPA and MTPO), formaldehyde (CH2O) and xylene (XYLE). Monoterpenes are primarily biogenic in origin, whereas

many of the remaining NMVOCs are known products of biomass burning, especially from peat fires (Yokelson et al., 1997). Beyond these, other emitted NMVOCs account for less than 5% of total OFP throughout the region.

We find that when OFP is sorted by the source sector of emitted NMVOCs, the majority of the OFP is attributed to biogenic NMVOCs, with only about 10% and 5% of the total OFP attributed each to pyrogenic and anthropogenic NMVOCs, respectively. However, even under VOC-limited conditions, $NO_x$ is still required for VOC oxidation to result in the production of ozone. Therefore, other sources of $NO_x$ can still play a role in ozone formation, even when ozone-forming NMVOCs are mainly emitted from the biogenic sector. We estimate the overlap of biogenic NMVOCs with anthropogenic and pyrogenic $NO_x$ by scaling OFP in each grid cell by the fraction of total $NO_x$ emissions that are attributed to each sector. The results are represented by the blue and orange hatching in Fig. 9b. According to our estimates, overlap between biogenic NMVOCs and anthropogenic $NO_x$ makes the largest contribution to OFP, accounting for 49% and 62% of monthly mean OFP in March and September, respectively. The overlap of biogenic NMVOCs with pyrogenic $NO_x$ comprises the next largest source of ozone, accounting for up to 24% of monthly mean OFP. With more than 90% of OFP from pyrogenic NMVOCs attributed to the same source of $NO_x$, we find overall that pyrogenic $NO_x$ accounts for 34% of total OFP in March of 2014 and 27% in September over Southeast Asia.

## 4.4 Uncertainties

All the methods we use to interpret observed variations of tropospheric ozone are imperfect in some way. Here, we summarise some of the prominent uncertainties associated with these methods.

One of the largest uncertainties relevant to this work is in the magnitude of biogenic NMVOC emissions. As mentioned in Sect. 2.3, the biogenic emissions model MEGAN2.1 has previously been shown to overestimate mean isoprene emissions in Southeast Asia by at least a factor of 2 (Langford et al., 2010; Stavrakou et al., 2014). Our model results suggest that the biogenic sector comprises the largest influence on OFP than any other NMVOC emission source and that isoprene is the single largest contributing NMVOC in Southeast Asia. Overestimation of biogenic NMVOCs in areas where ozone production is VOC-limited could explain positive bias in the model tropospheric ozone columns, especially in March when biogenic emissions are higher over a larger portion of the region than they are in September. Overestimating biogenic emissions might also affect the relative contribution of pyrogenic NMVOCs to total OFP: if the biogenic contribution is substantially less, the fraction attributable to pyrogenic NMVOCs might be higher, although we acknowledge that pyrogenic emissions are equally as uncertain as biogenic emissions.

Emissions of $NO_x$ provide another potentially significant source of uncertainty. Pyrogenic emissions of $NO_x$ from GFED3.1 were previously shown to be biased low over Southeast Asia by as much as 45% for the years 2005–2011 (Miyazaki et al., 2017). For the same time period, we find that GFED4.1s increases mean $NO_x$ emissions relative to GFED3 by 41%, which largely corrects this bias. Some of our anthropogenic inventories, however, are out of date for 2014, including the MIX inventory for Asia, which is only available through 2010 (Li et al., 2017). Between 2010 and 2014, anthropogenic emissions are thought to have increased in Southeast Asia (Kurokawa and Ohara, 2019), implying that our model simulations may underestimate anthropogenic sources of $NO_x$ and NMVOCs. Although we do not expect that this would affect the seasonal variability

in the anthropogenic emissions of either, our results suggest that the anthropogenic sector is a significant source of $NO_x$, which could still impact regional ozone production. If emissions of $NO_x$ are underestimated, from either anthropogenic or pyrogenic sources, ozone production could be higher and more VOC-limited than currently indicated by our results. This could also explain negative bias in the model tropospheric ozone columns, especially towards the end of the year when the anthropogenic
sector dominates the $NO_x$ emission distribution.

    Our analysis is further limited by assumptions made in the derivation and application of OFP. As mentioned in Sect. 4.3, one of the major assumptions inherent in the calculation of OFP is that conditions for ozone production must be VOC-limited. We have shown using mean daytime $L_N$/Q from the PBL that ozone production is generally predicted to be VOC-limited over areas where ozone production is highest. However, we have calculated OFP across the region, which includes many areas that
may be $NO_x$-limited or near the transition between regimes. As a result, our estimate of OFP likely constitutes an upper limit on the yield of ozone from emitted NMVOCs and should not be compared directly with P(O$_3$). Including areas that are not strictly VOC-limited may also inflate the influence of biogenic NMVOCs where $NO_x$ emissions are very low, for example over Borneo in March. As a sensitivity study, we recompute the sector-specific contributions to OFP after filtering by daytime PBL $L_N$/Q to exclude $NO_x$-limited conditions. This filtering procedure increases the pyrogenic NMVOC contribution from
about 10% in March and September to 39% and 63%, respectively. Although the relative contribution from biogenic NMVOCs decreases overall, a larger fraction overlaps with pyrogenic $NO_x$, increasing the total pyrogenic contribution from about 30% to 80% during both months. These values are much higher than those reported in Sect. 4.3 but also much more uncertain because we cannot be sure that our best estimate of $L_N$/Q precisely describes OFP, which is not similarly distributed throughout the atmosphere. Therefore, this treatment does not necessarily provide a better representation than is currently shown in Sect. 4.3,
but it does suggest that our reported estimate of the fractional pyrogenic contribution to OFP may be too low, perhaps by as much as a factor of 2–3.

# 5   Implications for public health

To assess the impact of regional ozone on public health, we use the maximum daily 8-hour average (MDA8) metric, which sets the basis for public health guidelines around the world. To calculate MDA8, we construct a 8-hour running mean of surface
ozone concentrations and then select the maximum value from each 24-hour day in local time. The World Health Organization (WHO) recommends a global limit on observed MDA8 ozone of 100 $\mu$g m$^{-3}$ ($\simeq$ 50 ppbv), based on evidence that exposure to ozone at this level increases the number of attributable deaths by 1–2% compared to a baseline of 70 $\mu$g m$^{-3}$ ($\simeq$35 ppbv) (WHO, 2005). Above the WHO recommended limit, the risk of mortality has been shown to increase with MDA8 ozone, even at short-term exposure timescales (Vicedo-Cabrera et al., 2020).
Mean MDA8 ozone exceeds the recommended exposure limit of 50 ppbv over a large fraction of Southeast Asia during both March and September of 2014. Figure 10 presents maps of monthly mean MDA8 ozone (ppbv), as calculated from the GEOS-Chem nested model. Excessive values of MDA8 generally occur over areas of biomass burning and are widespread across the region due to the relatively long lifetime of ozone, which can be a few days at the surface in Southeast Asia. During March,

prolonged exposure to ozone with MDA8 approaching 70 ppbv affects the majority of mainland Southeast Asia, home to 234 million people in 2014 (data.worldbank.org). During September, excessive ozone persists over Indonesia, Southeast Asia's single most populous country, with a population of more than 255 million people at that time. As a basis for comparison, we also include maps of MDA8 ozone for May and December of 2014, when emissions from biomass burning are at a minimum. During May, when biogenic emissions peak, MDA8 ozone approaches the WHO limit across the majority of the mainland but remains below 60 ppbv, and during December, MDA8 ozone is lower still and generally remains below 50 ppbv over land across the region. These results suggest that biomass burning coincides with the worst ozone air quality in the region, which exceeds WHO exposure guidelines.

In a recent study, Vicedo-Cabrera et al. (2020) analyzed short-term ozone-related deaths across 406 cities in 20 countries between 1985 and 2015. They found that exposure to surface ozone at levels above the WHO guideline was associated with a 0.2% mean increase in total short-term mortality. When we apply this factor to the gridded UN-adjusted population count product v4.11 for 2015 from the NASA Socioeconomic Data and Applications Center (sedac.ciesin.columbia.edu), scaled to 2014 using national mortality and population data from the World Bank (data.worldbank.org), we estimate that excessive ozone from biomass burning caused nearly 260 and 160 excess deaths across Southeast Asia in March and September, respectively. The mortality rate from Vicedo-Cabrera et al. varied substantially across the sampled countries, and did not include any countries from Southeast Asia. Furthermore, their study did not account for interannual or seasonal variability, which we have shown exerts a large influence within our study domain. As a result, our estimate for the number of excess deaths in Southeast Asia is quite uncertain. However, our estimate is not unreasonable compared to Marlier et al. (2013), who estimated that severe fires in Indonesia pushed surface ozone concentrations above the WHO limit for 150 days in 1997, causing 4,100 excess premature deaths. According to GFED4.1s fire emissions were more than 3 times higher in 1997 than they were in 2014, and they caused 2.5 times more exceedance days than are examined in this work. If the estimate from Marlier et al. (2013) is scaled for conditions in 2014, biomass burning should account for about 550 total ozone-attributed deaths during our study period, which agrees reasonably well with our estimate above.

We evaluate our GEOS-Chem values for MDA8 ozone using ground observations from Malaysia, which experiences reduced air quality during both regional burning seasons. As of 2020, the New Ambient Air Quality Standard (NAAQS) limits MDA8 ozone in Malaysia to 100 $\mu$g m$^{-3}$, in line with WHO recommendations. The prior limit was set by the Malaysian Ambient Air Quality Guidelines (MAAQG), which restricted MDA8 ozone to 120 $\mu$g m$^{-3}$ (60 ppbv) and had been in place since 1989 (DOE, 2015). Figure 11 shows model and observed monthly mean MDA8 ozone (ppbv) at ground site locations across Malaysia for March and September of 2014. In March, the model predictions reproduce the observations reasonably well (r = 0.82; slope = 0.78, y-intercept = 12.79), with the highest values occurring along the western coast of peninsular Malaysia. This part of the country sits opposite the Riau province in Indonesia, which is a hotspot for fire emissions during the early burning season. The normalized mean bias (NMB) of the model, calculated as the mean difference between the model and observations, normalized by the mean of the observations, is about 10%. The model also accurately reproduces observed exceedances of monthly mean MDA8 ozone at 11 different ground sites relative to WHO guidelines, with two of these at the MAAQG level. However, the small positive model bias results in false MDA8 exceedances at another seven sites.

Model performance is worse in September (r = 0.79; slope = 0.77; y-intercept = 16.76), when the NMB increases to 22%. Model MDA8 ozone exceeds WHO guidelines at 17 ground sites, while only four of these are confirmed by observations. Although our analysis indicates that GEOS-Chem generally underestimates tropospheric ozone across Southeast Asia, the spatial distribution of the model bias shown in Figure 7f suggests that the model does overestimate tropospheric ozone over Malaysia in September, though the data coverage is insufficient to determine by how much. Although some ground sites are located near cities where anthropogenic sources may exert a larger influence on ozone production, we find that exceedances of monthly mean MDA8 are not confirmed by observations anywhere in Malaysia in either May or December, when regional biomass burning is at a minimum. Despite a positive model bias, these results suggest that, although biogenic and anthropogenic ozone precursors may form the majority of MDA8 ozone across Southeast Asia, it is with the added influence from biomass burning that ozone exposure is elevated to hazardous levels exceeding global and national guidelines.

## 6 Concluding remarks

We used the GEOS-Chem atmospheric chemistry model, in combination with satellite and surface observations, to investigate the seasonal impacts of biomass burning on ozone air quality throughout Southeast Asia in 2014. Analysis of emission inventories emphasized two distinct biomass burning regimes: 1) burning on the peninsular mainland that peaks in March and 2) burning in Indonesia that peaks in September. Although the two burning seasons differ in the type of vegetation burned, both emit $NO_x$ and VOCs, which are ozone precursors. We found that outside of burning areas, the underlying photochemical environment is generally $NO_x$-limited, dominated by anthropogenic $NO_x$ and biogenic NMVOC emissions. Overall, however, pyrogenic NMVOCs are responsible for about 10% of the regional OFP during both biomass burning seasons. Pyrogenic $NO_x$ is even more influential, shifting towards more VOC-limited ozone production and increasing the total pyrogenic contribution to about 30% of OFP. Using model output, we showed that peak biomass burning activity coincides with widespread ozone exposure at levels that exceed world public health guidelines. We estimated that excess ozone from biomass burning caused about 260 premature deaths across Southeast Asia in March of 2014 and another 160 deaths in September. Although the model tends to overestimate ozone near areas of biomass burning, surface observations confirm that hazardous ozone levels coincide with fire activity.

We found that model tropospheric ozone agrees relatively well with satellite (annual mean bias = −11%) and surface observations (r ≥ 0.79, NMB ≤ 22.23%), though some discrepancies remain unexplained. We suspect that these discrepancies are largely due to uncertainties associated with emission estimates. Biogenic emissions, for example, are calculated online in GEOS-Chem and may be overestimated in Southeast Asia by more than a factor of 2 (Langford et al., 2010; Stavrakou et al., 2014). The most recent inventory for anthropogenic emissions from Asia is from 2010, which is four years out of date with respect to 2014. Even biomass burning emissions, which are specified for 2014, vary greatly among the four inventories that are available for use within GEOS-Chem (Bauwens et al., 2016; Miyazaki et al., 2017). We also acknowledge that model uncertainties are not only limited to emission inventories, but may be introduced through other components as well, including chemistry, mixing, deposition, and transport (Travis et al., 2016; Eastham and Jacob, 2017; Marvin et al., 2017; Silva and

Heald, 2018). Such uncertainties not only affect model ozone concentrations, but also the sensitivity of model ozone to its precursors and the influence of different source sectors. Development of effective air quality policy requires knowledge of how well these models describe surface air pollution and its driving processes.

Studies like ours would benefit immensely from further investigation into precursor emissions and the underlying chemical and physical processes that determine ozone across Southeast Asia. A regional network of ground-based remote sensing instruments for ozone and its precursors (e.g., Herman et al., 2009) would provide complementary measurements and serve as a useful validation source for satellite retrievals. Progress in space-borne sensor technology has improved our ability to observe spatial scales associated with Southeast Asia. Improved temporal coverage will come from sensors aboard geostationary

satellites that effectively hover over a particular geographical region. The South Korean Geostationary Environmental Satellite (GEMS), launched in February 2020, heralds the availability of multiple measurements of tropospheric ozone (Bak et al., 2013) and its precursors (Hong et al., 2017; Kwon et al., 2017) per day over Southeast Asia. Integration of these atmospheric data will ultimately require data assimilation or an inverse method framework (e.g., Miyazaki et al., 2020).

*Code and data availability.*  Model code and input data are free and available from the GEOS-Chem website (www.geos-chem.org). The

model website also links to the source databases from which input is prepared, including GFED (http://globalfiredata.org).

*Author contributions.*  MRM and PIP designed experiments and wrote the paper. BGL, RS and BJK provided ozone retrieval data. MTL and MFK provided access to surface air quality data. All co-authors assisted in revising the paper.

*Competing interests.*  The authors declare that they have no conflict of interest.

*Acknowledgements.*  Many thanks to the Department of Environment, Ministry of Environment and Water of Malaysia, for providing surface

air quality data. This study was funded as part of NERC's support of the National Centre for Earth Observation: MRM, PIP, BGL, RS and BJK were supported by grant number #NE/R016518/1, with further support for MRM from grant number #NE/R000115/1. Production of OMI height-resolved data at Rutherford Appleton Laboratory was funded through ESA's Climate Change Initiative.

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

**Table 1.** Emitted VOCs in GEOS-Chem with the highest assigned MIR (g ozone formed per g VOC emitted). Adapted from Carter (2010).

| VOC Species | GC Name | MIR | VOC Species | GC Name | MIR |
|---|---|---|---|---|---|
| Propene | PRPE | 11.66 | Acetaldehyde | ALD2 | 6.54 |
| Isoprene | ISOP | 10.61 | Methacrolein | MACR | 6.01 |
| Formaldehyde | CH2O | 9.46 | Limonene | LIMO | 4.55 |
| Xylene | XYLE | 7.84 | Monoterpenes | MTRP | 4.04 |
| Aldehydes $\geq$ C3 | RCHO | 7.08 | Toluene | TOLU | 4.00 |

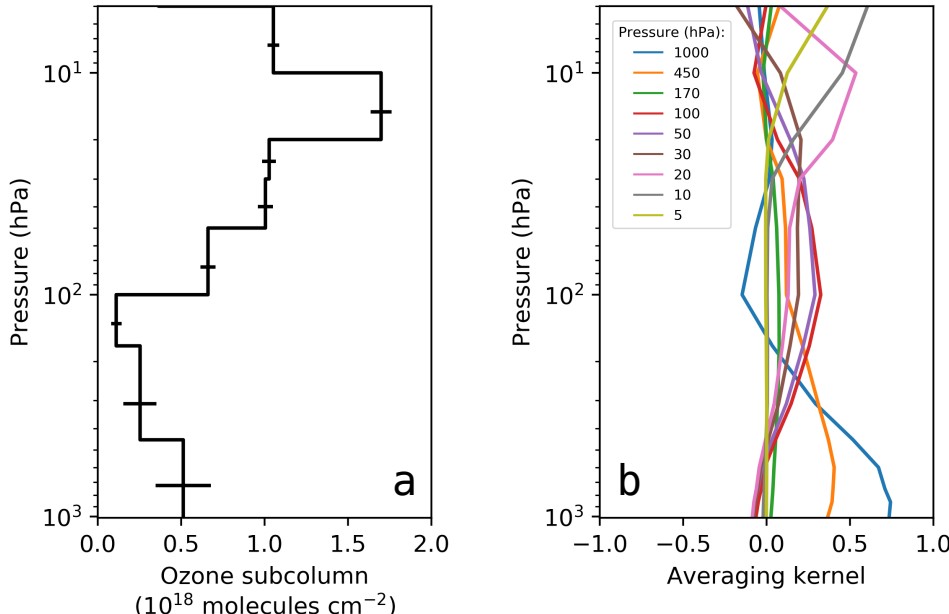

**Figure 1.** Example OMI ozone profile up to 5 hPa retrieved over mainland Southeast Asia (lat = 15.45° N, lon = 103.91° E) at the local satellite overpass time (solar zenith angle = 28.79°) on 4 March 2014. (a) Ozone subcolumns and retrieval error ($10^{18}$ molec cm$^{-2}$). (b) Averaging kernel matrix. Indicated pressures correspond to layer bottoms.

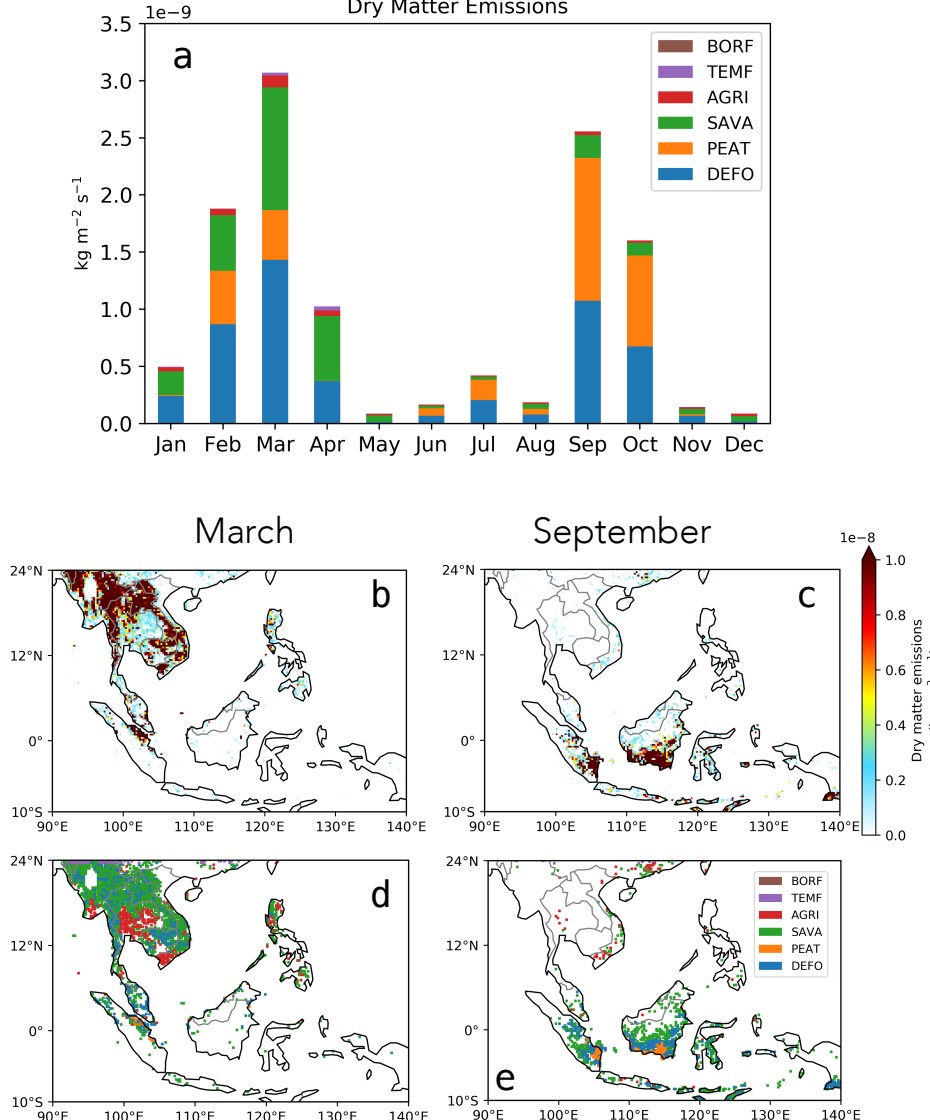

**Figure 2.** Monthly dry matter emissions (kg m$^{-2}$ s$^{-1}$) for Southeast Asia in 2014, as estimated by the GFED4.1s biomass burning inventory. (a) Timeseries of dry matter emissions. (b,c) Spatial distribution of dry matter emissions in March and September. (d, e) Map of the predominant vegetation types burned in March and September. Vegetation types shown in (a, d, e) include deforestation (DEFO), peat (PEAT), savanna (SAVA), agricultural waste (AGRI) and temperate forest (TEMF). There are no emissions from boreal forests (BORF) in this region.

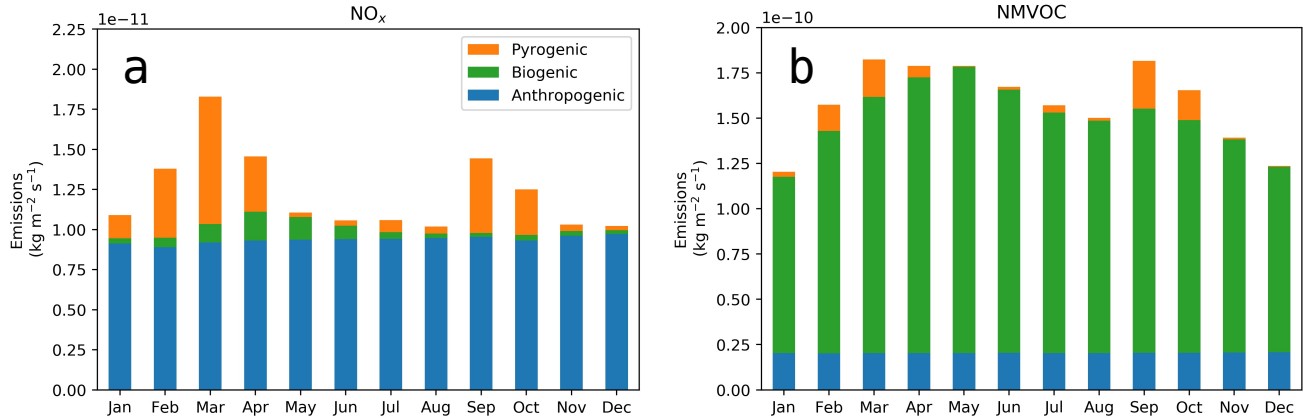

**Figure 3.** Timeseries of monthly mean (a) $NO_x$ and (b) NMVOC emissions (kg m$^{-2}$ s$^{-1}$) from Southeast Asia in 2014. Emissions are sorted by source sector: anthropogenic, biogenic or pyrogenic. Source attribution for the full timeseries is extracted from the $2° × 2.5°$ global simulation.

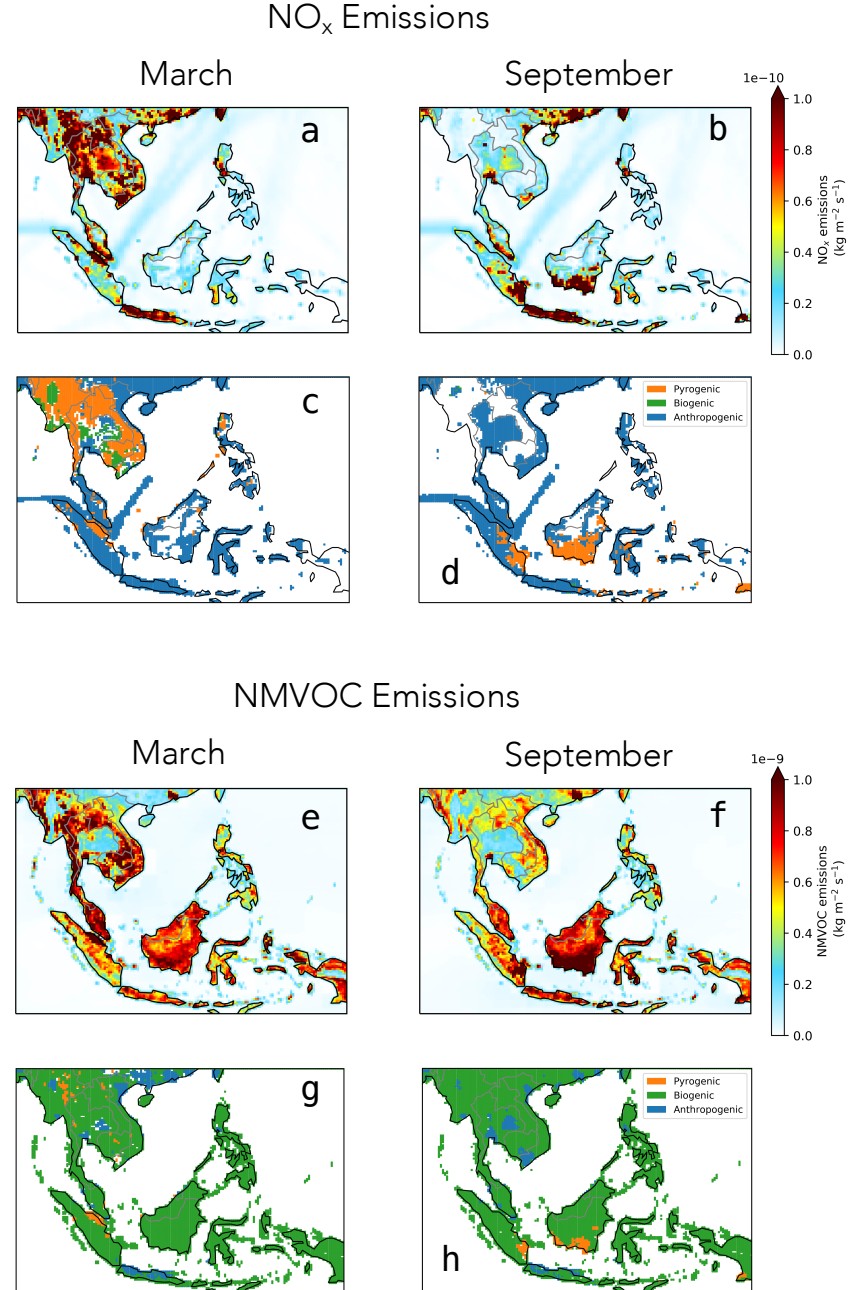

**Figure 4.** Monthly mean emissions (kg m$^{-2}$ s$^{-1}$) of NO$_x$ and NMVOCs across Southeast Asia in March and September of 2014. (a, b, e, f) Spatial distribution of emissions. (c, d, g, h) Map of the predominant emission source types: anthropogenic, biogenic or pyrogenic.

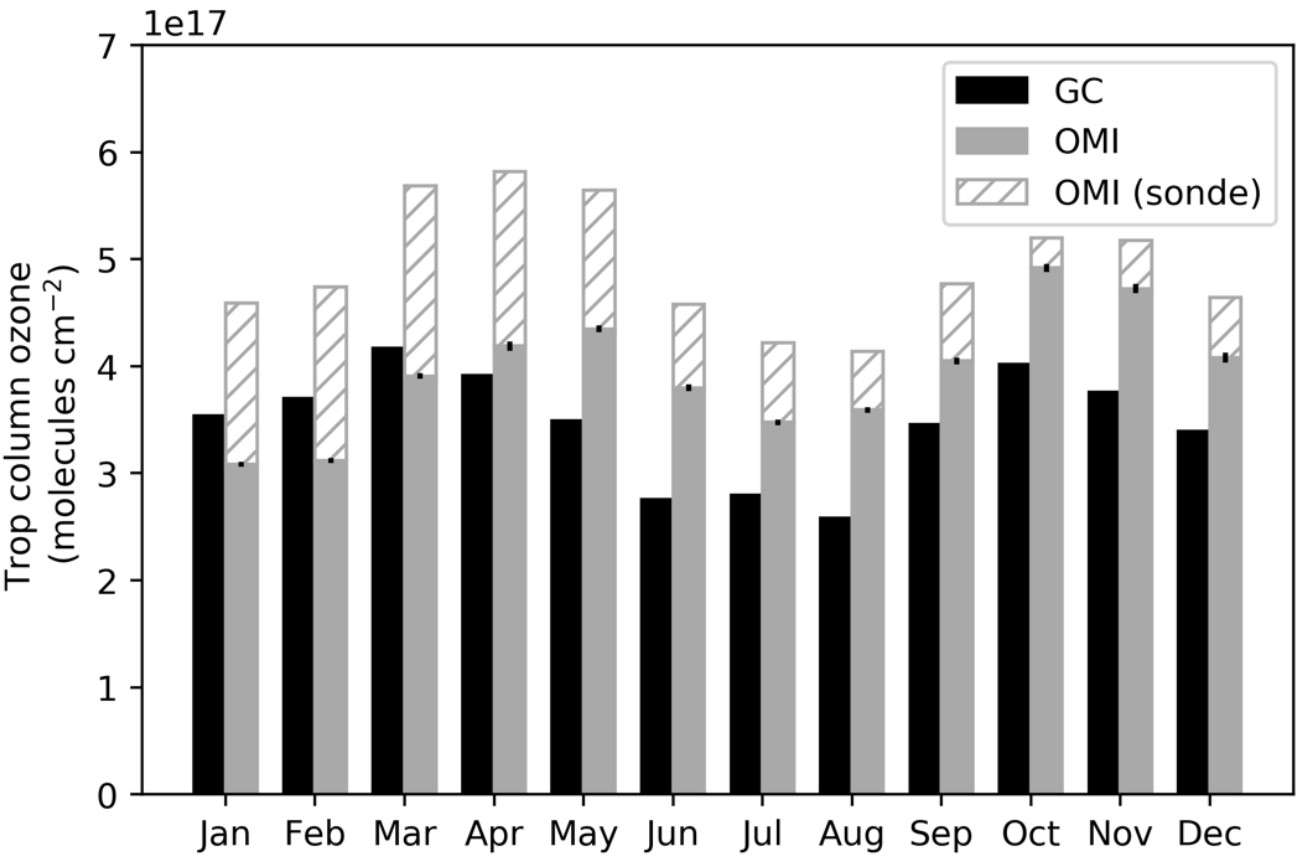

**Figure 5.** Timeseries of GEOS-Chem model and OMI monthly mean tropospheric ozone column (molec cm$^{-2}$) across Southeast Asia in 2014. Model results for the full timeseries are from the $2° \times 2.5°$ global simulation. Error bars shown for OMI are the standard-errors-in-the-mean, as indication of random error. Also shown are OMI values adjusted for bias estimates with respect to an ozonesonde ensemble, as indication of the systematic error range.

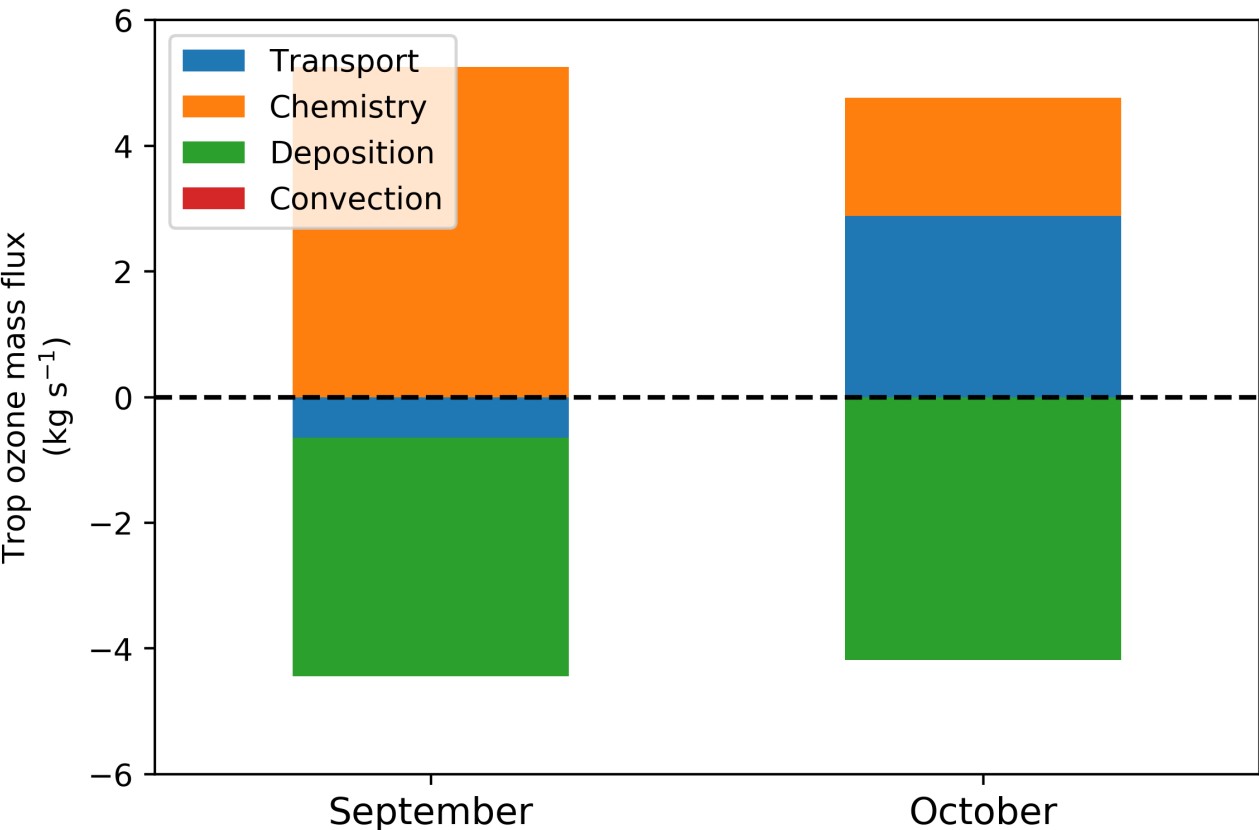

**Figure 6.** GEOS-Chem model mean tropospheric ozone mass fluxes (kg s$^{-1}$) over Southeast Asia in September and October of 2014, averaged on the $2° \times 2.5°$ global model grid. Fluxes are computed within GEOS-Chem and represent the difference in the model column mass before and after each component is applied. The troposphere is defined in this simulation using the dynamic tropopause height from GEOS-FP, which may not precisely correspond to the tropospheric columns defined in Sect. 2.1.

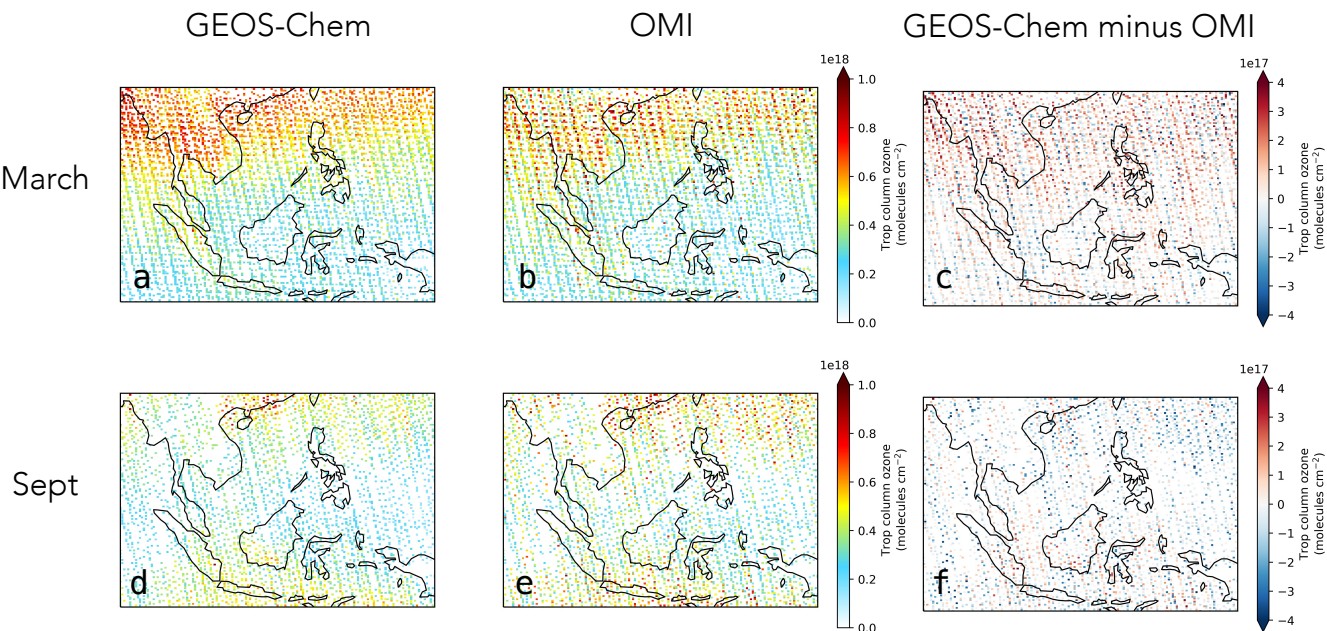

**Figure 7.** Spatial distribution of (a, d) GEOS-Chem model and (b, e) OMI monthly mean tropospheric ozone column (molec cm$^{-2}$) across Southeast Asia in March and September of 2014. Panels c and f show the difference between GEOS-Chem and OMI for March and September, respectively. Ozone columns are shown on the $0.25° \times 0.3125°$ nested grid and are represented by white space where OMI observations are not available.

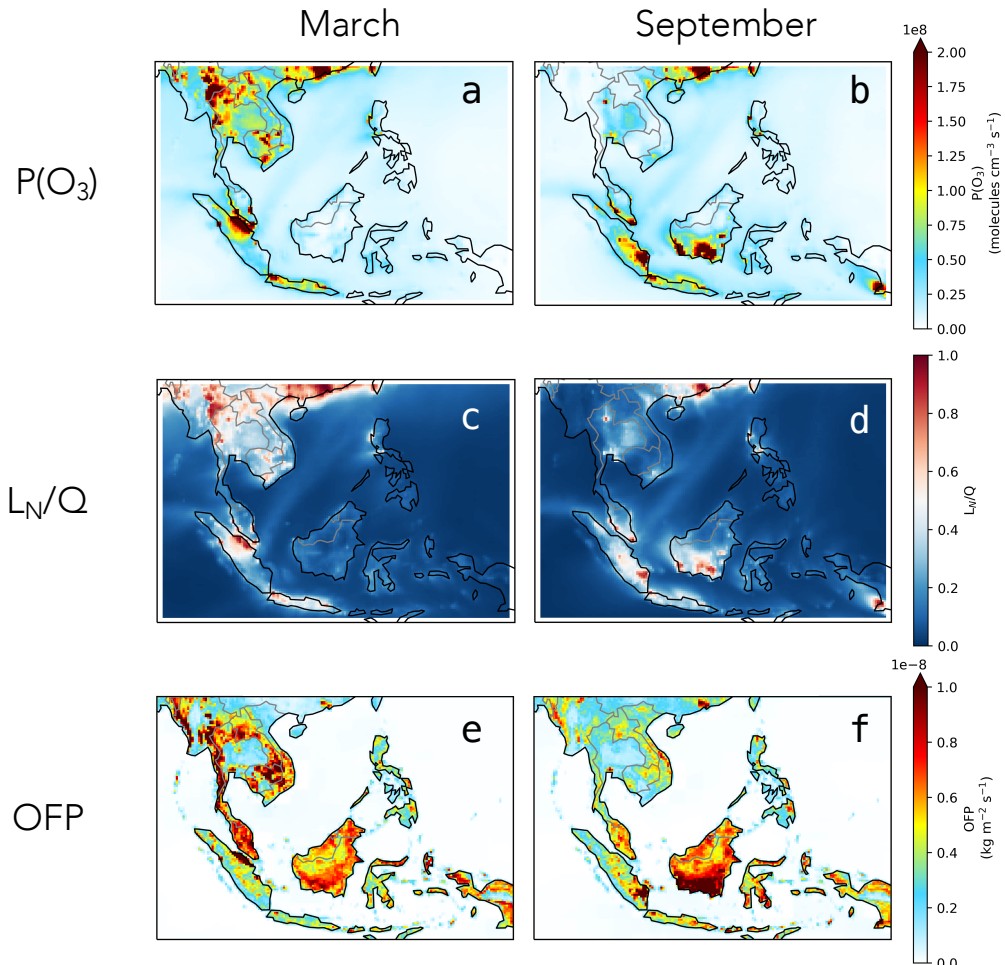

**Figure 8.** Spatial distribution of model ozone sensitivity indicators across Southeast Asia in March and September of 2014. (a, b) Mean ozone production rate (molec cm$^{-3}$ s$^{-1}$) in the PBL during the daytime (0600–1800). (c, d) Mean L$_N$/Q in the PBL during the daytime (0600–1800). (e, f) Ozone formation potential (kg m$^{-2}$ s$^{-1}$).

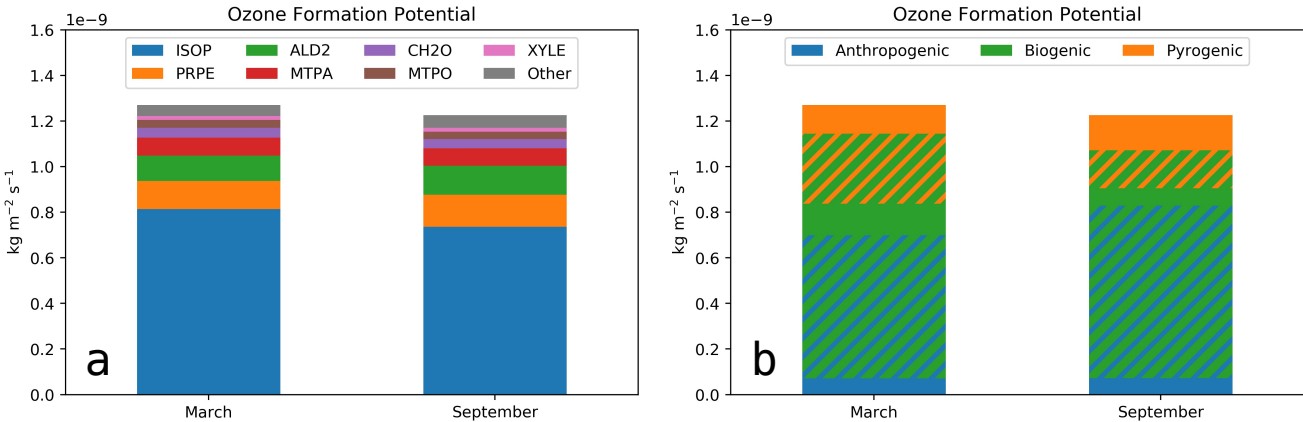

**Figure 9.** Mean ozone formation potential (kg m$^{-2}$ s$^{-1}$) for Southeast Asia in March and September of 2014 sorted by (a) individual species of emitted NMVOCs and (b) source sector of emitted NMVOCs. Blue and orange hatching shows OFP from biogenic NMVOCs, scaled by the fraction of NO$_x$ emissions that are attributed to anthropogenic and pyrogenic sources, respectively.

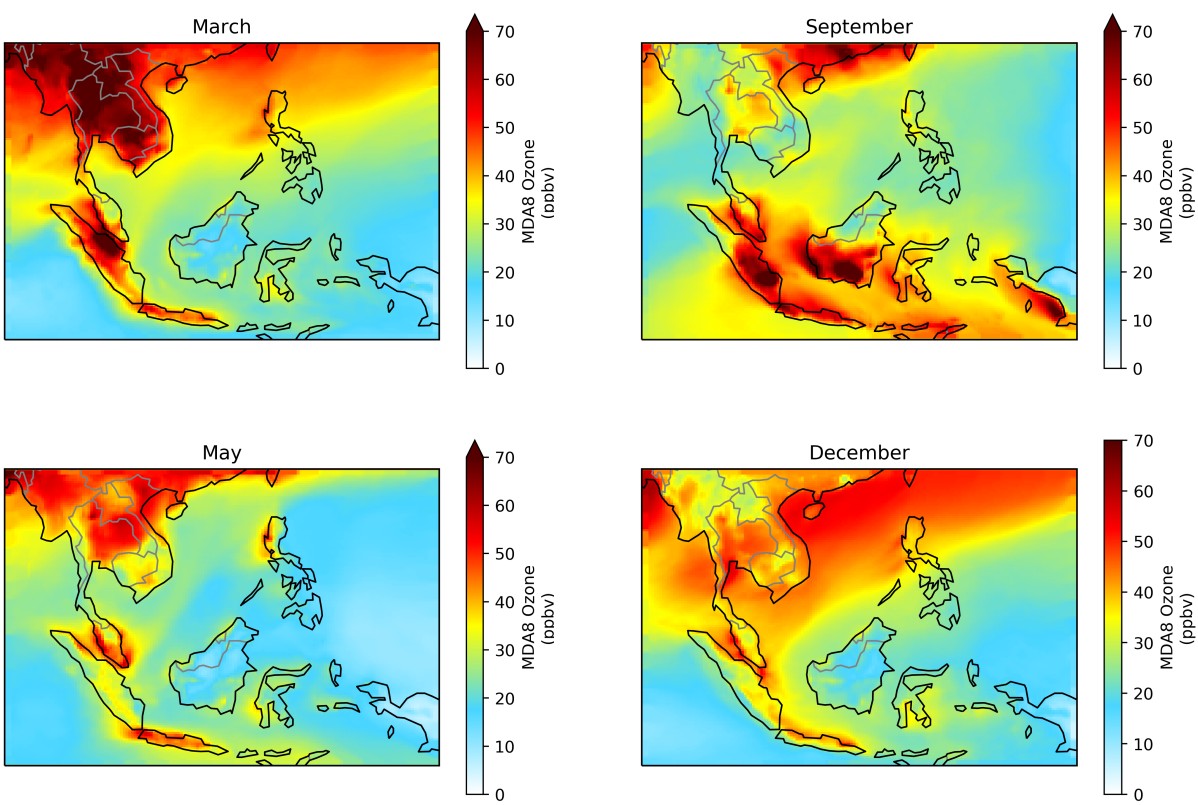

**Figure 10.** Spatial distribution of monthly mean MDA8 ozone (ppbv) calculated for Southeast Asia in March, May, September and December of 2014, based on surface concentrations of ozone from the GEOS-Chem nested model. For reference, the World Health Organization recommends an observed limit of $\simeq 50$ ppbv.

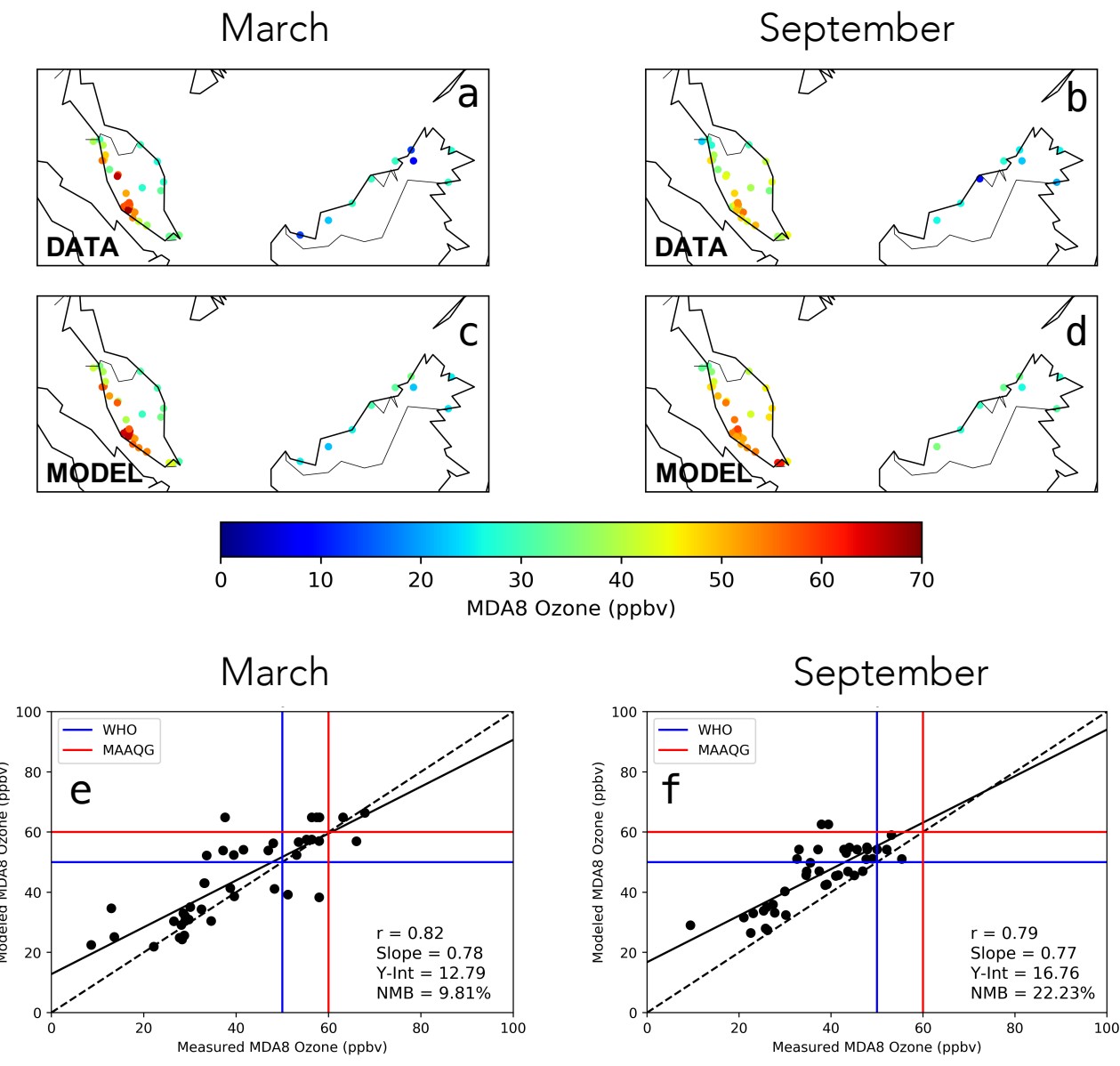

**Figure 11.** Monthly mean MDA8 ozone (ppbv) across Malaysia in March and September of 2014. (a–d) Observed and GEOS-Chem spatial distribution of MDA8 ozone at air quality monitoring sites. (e, f) Scatterplot of observed and GEOS-Chem MDA8 ozone at the air quality sites. A line of best fit is shown in black, with the associated mean statistics shown inset: Pearson correlation (r), the slope and y-intercept of the best-fit line, and the normalized mean bias (NMB) as defined in the main text. Blue and red lines denote WHO and MAAQG MDA8 limits as of 2014.