# Peer review of "Photochemical environment over Southeast Asia primed for hazardous ozone levels with influx of nitrogen oxides from seasonal biomass burning"

_Atmospheric Chemistry and Physics, 2020_

## Referee Comment (RC1) · Anonymous Referee #1 · 18 Nov 2020

Recommendation: Return to author for minor revisions Journal: ACP Title: Photochemical environment over Southeast Asia primed for hazardous ozone levels with influx of nitrogen oxides from seasonal biomass burning Author(s): Margaret R. Marvin et al. MS No.: acp-2020-886

General comments The study explores impacts of seasonal biomass burning on ozone production using GEOS-Chem along with the OMI data and the emission and fire inventories. The fire inventories reveal two interesting biomass burning regimes. Three indices derived from the GEOS-Chem model are used to investigate the mechanisms

of ozone formation with a focus on the two seasonal biomass burning regimes. Uncertainties associated with the inventories and assumptions are properly discussed. The presented results show interesting findings in terms of anthropogenic and biogenic emission interaction and its impact on air quality. If there is a weakness, it is my opinion that it relates to the calculation and interpretation of the indices. I believe this concern can be addressed through moderate revision.

Specific comments 1.-Ozone production rates. Can you explain what RO2 are included in Equation 1?

2.-Ozone sensitivity to NOx and VOCs. The organic nitrates can be important in an isoprene-dominated area under high NOx conditions. Can you roughly estimate the impact of organic nitrates? In addition, the organic peroxide can also be a sink for HOx radicals. Can you also quantify the contribution from the organic peroxides if possible? Bates et al. (2019) updated the isoprene mechanism in GEOS-Chem, which includes important updates on isoprene nitrates. It might be helpful to test it with the new isoprene chemistry, but probably not essential for this study.

3.-Ozone formation potential. The authors did not filter the OFP for potential NO-limited regions and acknowledged the associated uncertainties. In Figure 9, how different it would be if you exclude the potential NO-limited regions?

4.-PBL ozone. PBL-averaged values of the three indices are used here. However, the simulated PBL-averaged O3 is not justified, as OMI provide tropospheric O3 and the ground measurements do not seem to cover the two regions of biomass burning. Is there a way to justify that the PBL O3 is well represented by the model in the two regimes of interest?

5.-The three indices are focused on the chemistry of ozone production in the PBL. However, other physical processes could be important for PBL O3 too as pointed out by the authors in the last Section. Figure 6 shows a glimpse of those processes for tropospheric O3. I wonder their relative importance to chemistry for PBL O3. To be

specific, I know convection is an important source of surface O3 in Amazonian region due to the active MSC. Therefore, I wonder if that's true for this tropical region too and how it could potentially affect the results of O3 chemistry in this study.

Minor comments 1.-Can you add latitudes & longitude for Figure 1b-e

2.-Line 212. Can you clarify the source region of the transport?

---

## Referee Comment (RC2) · Anonymous Referee #2 · 19 Nov 2020

This manuscript investigated the impacts of biomass burning on the air quality especially ozone pollution in Southeast Asia. Using GOES-Chem model as well as satellite observations, the authors found that VOC-limited ozone production exists in the burning areas, while outside the burning areas NOx-limited regime dominates. This study also estimated the impacts on public health due to the biomass burning emission.

The text is concisely written and well documented. The topic is applicable for the Atmospheric Chemistry & Physics journal. However, the current manuscript lacked detailed discussion and needed more analysis (please see the remarks below). My

main concern is, the tropospheric ozone pollution is determined by multiple factors such as surface temperature, precipitation, and emissions (anthropogenic, biogenic, and pyrogenic for Southeast Asia). The current manuscript identified two major episodes when pyrogenic emissions could play an important role in two regions, i.e., peninsular mainland in March and Indonesia in September. As discussed in the remarks below, the high ozone concnetrations are found not only in and near the biomass burning areas, but also in other regional far away from the fire activities. So it is difficult to conclude that the high ozone events are caused by the pyrogenic emissions. One sensitivity experiment without GFED emissions is necessary to simulate the ozone pollution without biomass burning, and the differences between these two GEOS-Chem runs can better demonstrate the impacts of fire activities.

In summary, the current manuscript shows important results but need further work. Major revisions as indicated in the comments and remarks below are needed before consideration of publication in ACP.

Detailed Remarks/Suggestions for Revision

Line 206: I am confused about the Figure 5 and the bias correction with respect to an ozonesonde ensemble. My understanding is that the authors applied the OMI averaging kernels (AK) to sonde profiles to calculate 'profiles' retrieved by OMI. Te GEOS-Chem model should generate true concentration profiles, so I don't think it is proper to compares these two. Should the OMI AKs be applied to the GEOS-Chem profiles and them be compared with OMI retrievals?

Line 213: Any details about how the flux especially transport is calculated? Through the lateral boundaries of the nested high resolution GEOS-Chem domain?

Line 214-218: Is the 'mean annual' value calculated through averaging the domain? Can the authors show the annual cycle in a table or a figure?

Line 219-225: From Figure 7, it is hard to conclude that the GC simulated high ozone

columns in the high latitude, i.e., peninsular mainland, are correlated to the biomass burning. For instance, over the ocean area between Hainan Island and Taiwan, there are also high ozone columns in both GC simulations and OMI observations in March. To me, the high ozone columns in the high latitude are more regional nature of chemistry and climate, and cannot be explained by the biomass burning. If the biomass burning dominates, it cannot explain why oceans in both side of the peninsular mainland have high ozone when only one side is the downwind regional of the biomass burning. The Sept case has the similar results. GC did not simulate the high ozone columns over or near the locations of biomass burning (Fig. 2d). Lastly, the difference plots show mixed high and low bias, both of which are large. Does it mean GC has problem to simulate the ozone columns in this region? A control run without GEFD emissions may be helpful here to identify the contribution from biomass burning.

Line 373: What kind of population data were used in this study? Since the GEOS-Chem simulated concentrations at gridded domain, it may be more proper to apply the gridded population products such as the gridded NASA SEDAC global population products (https://sedac.ciesin.columbia.edu/data/collection/gpw-v4).

Line 387: Is possible to add the GC simulated ozone concentrations as background in a) and b)? Hard to tell if GC capture the pattern of ozone when comparing Figures 10 and 11 which do not have the same map ratio.

Figures

Figure 2. '(b-e)' should be '(b-c)' for Spatial distribution of dry matter emissions?

Figure 7. What are the resolution of GC and OMI? If they are gridded, why not show contours? The current plots look like there are lots of gaps.

[Figure]

---

## Author Comment (AC1) · 17 Dec 2020

**Response to Referee Comments**

Corresponding author: Margaret R. Marvin (mmarvin@ed.ac.uk)

Many thanks to our referees for taking the time to provide thoughtful comments on our manuscript. Here we respond to referee comments, with comments shown in bold and our responses in plain text. Manuscript passages that have been revised in response to referee comments are shown italicized.

**Response to Referee #1**

**The study explores impacts of seasonal biomass burning on ozone production using GEOS-Chem along with the OMI data and the emission and fire inventories. The fire inventories reveal two interesting biomass burning regimes. Three indices derived from the GEOS-Chem model are used to investigate the mechanisms of ozone formation with a focus on the two seasonal biomass burning regimes. Uncertainties associated with the inventories and assumptions are properly discussed. The presented results show interesting findings in terms of anthropogenic and biogenic emission interaction and its impact on air quality. If there is a weakness, it is my opinion that it relates to the calculation and interpretation of the indices. I believe this concern can be addressed through moderate revision.**

**Specific comments:**
**1.-Ozone production rates. Can you explain what RO2 are included in Equation 1?**

Equation 1 is provided to demonstrate how ozone production relates to [NO], [$HO_2$] and [$RO_2$]. However, we do not calculate ozone production ourselves but rather utilize ozone production rates output by GEOS-Chem, which are calculated within the model as the sum of the rates for all chemical reactions that produce ozone. This accounts for all $RO_2$ that produce ozone within the 'complexSOA_SVPOA' mechanism, including $RO_2$ derived from isoprene, monoterpenes and smaller VOCs. To clarify that we use model-computed ozone production rates and to explain which $RO_2$ are represented by this method, we have updated the text at Line 242 as follows:

*"The subscript i denotes that the second term is summed over multiple species of organic peroxy radical. We use ozone production rates calculated by GEOS-Chem, which accounts for all contributing species as represented here by the 'complexSOA_SVPOA' mechanism, including $RO_2$ derived from isoprene, monoterpenes and smaller VOCs."*

**2.-Ozone sensitivity to NOx and VOCs. The organic nitrates can be important in an isoprene-dominated area under high NOx conditions. Can you roughly estimate the impact of organic nitrates? In addition, the organic peroxide can also be a sink for HOx radicals. Can you also quantify the contribution from the organic peroxides if possible? Bates et al. (2019) updated the isoprene mechanism in GEOS-Chem, which includes important updates on isoprene nitrates. It might be helpful to test it with the new isoprene chemistry, but probably not essential for this study.**

In the 'complexSOA_SVPOA' mechanism, the fate of organic nitrates if not otherwise returned to $NO_2$ is to undergo hydrolysis to form either $HNO_3$ or INDIOL, an aerosol species that is dominated by organic nitrates derived from isoprene and monoterpenes. Using model output from the nested simulation, we find that P(INDIOL) represents an additional sink for $NO_x$ at about 15% of the value of total P($HNO_3$) on average across Southeast Asia during the daytime (1330) in both March and September of 2014. An equivalent calculation is not available for organic peroxides because many of these species participate in subsequent chemistry that may result in the recycling of $HO_x$. We agree that competing loss processes for $HO_x$ are important to the calculation of $L_N/Q$, but this is difficult to investigate using GEOS-Chem and would likely require box modeling support. Given these challenges, it is unlikely that additional simulations with the updated chemistry from Bates and Jacob (2019) will significantly change our calculation for $L_N/Q$ at this time. However, we have updated the text at Line 264 to address the potential influence of organic nitrates and peroxides:

*"This method of calculating $L_N/Q$ assumes a simplified removal scheme for $HO_x$ and $NO_x$ radicals, as both can also be removed through the formation of organic products. Loss of isoprene nitrates to hydrolysis has been shown to provide an important sink for $NO_x$ (Bates and Jacob, 2019). Although the organic sink for $HO_x$ may offset the impact of organic nitrates, formation of organic peroxides is intertwined with complex recycling mechanisms, which makes it difficult to assess the total impact of these uncertainties on our calculation of $L_N/Q$ at this time."*

Bates, K. H. and Jacob, D. J.: A new model mechanism for atmospheric oxidation of isoprene: global effects on oxidants, nitrogen oxides, organic products, and secondary organic aerosol, *Atmos. Chem. Phys.*, 19, 9613–9640, https://doi.org/10.5194/acp-19-9613-2019, 2019.

**3.-Ozone formation potential. The authors did not filter the OFP for potential NO-limited regions and acknowledged the associated uncertainties. In Figure 9, how different it would be if you exclude the potential NO-limited regions?**

We have replotted Fig. 9 here after filtering OFP to exclude grid cells that are indicated by daytime PBL $L_N/Q$ to be $NO_x$-limited:

[Figure]

This filtering procedure increases the pyrogenic NMVOC contribution from ~10% in March and September to 39% and 63%, respectively. Although the relative contribution from biogenic NMVOCs decreases overall, a larger fraction overlaps with pyrogenic $NO_x$, increasing the total pyrogenic contribution from ~30% to ~80% during both months. These values are much higher than we currently report but also highly uncertain because we cannot be sure that our best estimate of $L_N/Q$, which varies greatly with time and 3-D space, precisely describes OFP, which is coarser with respect to time and essentially 2-D. Due to these uncertainties, we feel that this treatment does not necessarily provide a better representation than is currently shown in Fig. 9, but it does support the suggestion that our current figures likely underestimate the pyrogenic contribution to OFP, perhaps by as much as a factor of 2–3. We update the paragraph beginning at Line 340 to include a description of this sensitivity study and its results:

*"Our analysis is further limited by assumptions made in the derivation and application of OFP. As mentioned in Sect. 4.3, one of the major assumptions inherent in the calculation of OFP is that conditions for ozone production must be VOC-limited. We have shown using mean daytime $L_N/Q$ from the PBL that ozone production is generally predicted to be VOC-limited over areas where ozone production is highest. However, we have calculated OFP across the region, which includes many areas that may be $NO_x$-limited or near the transition between regimes. As a result, our estimate of OFP likely constitutes an upper limit on the yield of ozone from emitted NMVOCs and should not be compared directly with $P(O_3)$. Including areas that are not strictly VOC-limited may also inflate the influence of biogenic NMVOCs where $NO_x$ emissions are very low, for example over Borneo in March. As a sensitivity study, we recompute the sector-specific contributions to OFP after filtering by daytime PBL $L_N/Q$ to exclude $NO_x$-limited conditions. This filtering procedure increases the pyrogenic NMVOC contribution from about 10% in March and September to 39% and 63%, respectively. Although the relative contribution from biogenic NMVOCs decreases overall, a larger fraction overlaps with pyrogenic $NO_x$, increasing the total pyrogenic contribution from about 30% to 80% during both months. These values are much higher than those reported in Sect. 4.3 but also much more uncertain because we cannot be sure that our best estimate of $L_N/Q$ precisely describes OFP, which is not similarly distributed throughout the atmosphere. Therefore, this treatment does not necessarily provide a better representation than is currently shown in Sect. 4.3, but it does suggest that our reported estimate of the fractional pyrogenic contribution to OFP may be too low, perhaps by as much as a factor of 2–3."*

**4.-PBL ozone. PBL-averaged values of the three indices are used here. However, the simulated PBL-averaged O3 is not justified, as OMI provide tropospheric O3 and the ground measurements do not seem to cover the two regions of biomass burning. Is there a way to justify that the PBL O3 is well represented by the model in the two regimes of interest?**

The use of PBL-averaged values is intended to aid in the interpretation of ozone chemistry within the mixing layer where it is most likely to influence surface air quality. Due to a scarcity of measurements in our domain of interest, substantive datasets are not available to explicitly evaluate model PBL ozone. Instead, we evaluate regional ozone using ground data, where available, as well as tropospheric columns from OMI. As shown in Fig. 1b, the averaging kernel matrix used to describe the OMI columns has a finer vertical scale than the columns themselves and reaches its maximum sensitivity around 860 hPa, which approaches the mean daytime PBL for our model domain. Because the averaging kernel is applied to the model, this lends confidence that the PBL

is reasonably well represented when we evaluate the model ozone columns in Sect. 3.3. We have updated the text at Line 95 to clarify this point:

*"Figure 1b shows that the averaging kernel for the surface–450 hPa layer of the same profile is greater than 0.7 in that vertical range, with increasing sensitivity towards the daytime planetary boundary layer (PBL)."*

**5.-The three indices are focused on the chemistry of ozone production in the PBL. However, other physical processes could be important for PBL O3 too as pointed out by the authors in the last Section. Figure 6 shows a glimpse of those processes for tropospheric O3. I wonder their relative importance to chemistry for PBL O3. To be specific, I know convection is an important source of surface O3 in Amazonian region due to the active MSC. Therefore, I wonder if that's true for this tropical region too and how it could potentially affect the results of O3 chemistry in this study.**

The equivalent of Fig. 6 is shown here for PBL ozone in March and September of 2014:

[Figure]

The referee is correct that convection exerts a stronger influence on PBL ozone than is indicated by the tropospheric mass fluxes, but we find that the influence of convection on the total positive mass flux is small (~10% in both March and September) compared to the influence of chemistry (85% and 70%, respectively) in Southeast Asia. Although the physical ozone fluxes may affect air quality near the surface, they should not significantly impact the chemistry itself, which would be more sensitive to the processes affecting $NO_x$ and VOC precursors. The evaluation of model surface ozone in Sect. 5 should account for the majority of these processes. Further investigation requires work that is beyond the scope of this current manuscript but will provide a basis for future studies.

**Minor comments:**
 **1.-Can you add latitudes & longitude for Figure 1b-e**

Yes, this is now done.

**2.-Line 212. Can you clarify the source region of the transport?**

The mass fluxes shown in Fig. 6 are not horizontally directional and since we cannot confirm in which direction the transport is moving, we prefer not to clarify the source region of the transport at this time.

**The text is concisely written and well documented. The topic is applicable for the Atmospheric Chemistry & Physics journal. However, the current manuscript lacked detailed discussion and needed more analysis (please see the remarks below). My main concern is, the tropospheric ozone pollution is determined by multiple factors such as surface temperature, precipitation, and emissions (anthropogenic, biogenic, and pyrogenic for Southeast Asia). The current manuscript identified two major episodes when pyrogenic emissions could play an important role in two regions, i.e., peninsular mainland in March and Indonesia in September. As discussed in the remarks below, the high ozone concnetrations are found not only in and near the biomass burning areas, but also in other regional far away from the fire activities. So it is difficult to conclude that the high ozone events are caused by the pyrogenic emissions. One sensitivity experiment without GFED emissions is necessary to simulate the ozone pollution without biomass burning, and the differences between these two GEOS-Chem runs can better demonstrate the impacts of fire activities.**

**Detailed Remarks/Suggestions for Revision:**
**Line 206: I am confused about the Figure 5 and the bias correction with respect to an ozonesonde ensemble. My understanding is that the authors applied the OMI averaging kernels (AK) to sonde profiles to calculate 'profiles' retrieved by OMI. Te GEOS-Chem model should generate true concentration profiles, so I don't think it is proper to compares these two. Should the OMI AKs be applied to the GEOS-Chem profiles and them be compared with OMI retrievals?**

We apologize for the confusion. Our procedure is consistent with the referee's recommendation: we apply OMI AKs to the GEOS-Chem ozone profiles, which we then use to generate tropospheric columns that are comparable with the OMI measurements (Line 150 in Sect. 2.3). However, the AKs are also used to derive the ozonesonde bias correction, which represents the ground truth relative to the retrieval profile and is therefore applied to the OMI measurements (footnote on Page 7). The bias correction simply represents the systematic error range affecting the retrieved OMI columns.

To reiterate in Sect. 3.3 that the OMI AKs are applied appropriately to the GEOS-Chem profiles, we have added the following text at Line 209: *"Averaging kernels from OMI are applied to profiles from GEOS-Chem in order to produce comparable tropospheric ozone columns (Sect. 2.3)."*

**Line 213: Any details about how the flux especially transport is calculated? Through the lateral boundaries of the nested high resolution GEOS-Chem domain?**

The fluxes shown in Fig. 6 are output directly from GEOS-Chem and represent the difference in the model column mass before and after each component is applied. We show the mean mass fluxes extracted from the global $2° \times 2.5°$ simulation and averaged across the Southeast Asian domain. The transport term represents the mean horizontal flux computed for individual grid cells across the region. We have updated the figure caption to include more details about the flux calculations:

*"**Figure 6**. GEOS-Chem model mean tropospheric ozone mass fluxes (kg s⁻¹) over Southeast Asia in September and October of 2014, averaged on the 2∘ × 2.5∘ global model grid. Fluxes are computed within GEOS-Chem and represent the difference in the model column mass before and after each component is applied. The troposphere is defined in this simulation using the dynamic tropopause height from GEOS-FP, which may not precisely correspond to the tropospheric columns defined in Sect. 2.1."*

**Line 214-218: Is the 'mean annual' value calculated through averaging the domain? Can the authors show the annual cycle in a table or a figure?**

Yes, the mean annual bias is computed using the monthly mean tropospheric ozone columns shown in Fig. 5, which have been averaged across the Southeast Asian domain. To clarify that Fig. 5 shows domain-averaged values, we have updated the text at Line 206 to read: *"Figure 5 shows monthly mean GEOS-Chem model and OMI measurements of tropospheric ozone columns (molec cm⁻²) from 2014, averaged across Southeast Asia."* To clarify that the mean annual bias is computed using the mean values from Fig. 5, we have updated the text at Line 214 to read: *"Using the monthly mean tropospheric ozone columns from Fig. 5, we calculate that GEOS-Chem has a mean annual bias of –11% compared to OMI (up to –30% when OMI is adjusted for systematic error)."*

The annual cycle of the mean tropospheric ozone column over Southeast Asia, as modelled by GEOS-Chem and measured by OMI, is shown in Fig. 5.

**Line 219-225: From Figure 7, it is hard to conclude that the GC simulated high ozone columns in the high latitude, i.e., peninsular mainland, are correlated to the biomass burning. For instance, over the ocean area between Hainan Island and Taiwan, there are also high ozone columns in both GC simulations and OMI observations in March. To me, the high ozone columns in the high latitude are more regional nature of chemistry and climate, and cannot be explained by the biomass burning. If the biomass burning dominates, it cannot explain why oceans in both side of the peninsular main- land have high ozone when only one side is the downwind regional of the biomass burning. The Sept case has the similar results. GC did not simulate the high ozone columns over or near the locations of biomass burning (Fig. 2d). Lastly, the difference plots show mixed high and low bias, both of which are large. Does it mean GC has problem to simulate the ozone columns in this region? A control run without GEFD emissions may be helpful here to identify the contribution from biomass burning.**

We agree with the referee that it is difficult to conclude from Fig. 7 whether modeled ozone columns are correlated with biomass burning. Therefore, we are careful not to draw major conclusions from the spatial patterns alone, merely noting that we tend to see ozone enhancements over regions of biomass burning. Instead, we identify the contribution from biomass burning using a variety of ozone indicators, as described in Sect. 4. These methods account for the influence of local emissions and chemistry on regional ozone production without the need for a zero-emissions simulation. Running the model without GFED emissions would result in a non-linear chemical response that may not be directly comparable with the original simulation, which is why we use the indicators instead. Our results show that the majority of regional ozone production occurs over

areas of biomass burning (Fig. 8, panels a and b). The areas over ocean mentioned by the referee have much lower local ozone production, which suggests that these are instead primarily influenced by transport.

The difference plots in Fig. 7 do indeed show mixed high and low bias for GEOS-Chem, relative to OMI. We attribute this bias to model uncertainties, as discussed in Sect. 4.4. Since many of these uncertainties are associated with components beyond the influence of biomass burning, it is not likely that a control run without GFED emissions will correct this bias.

**Line 373: What kind of population data were used in this study? Since the GEOS- Chem simulated concentrations at gridded domain, it may be more proper to apply the gridded population products such as the gridded NASA SEDAC global population products (https://sedac.ciesin.columbia.edu/data/collection/gpw-v4).**

We used national population and mortality data from 2014 as provided by the World Bank (data.worldbank.org), but we agree that a gridded population product would be ideal for interpreting excessive model ozone values. We have therefore recalculated ozone-attributable deaths using the gridded UN-adjusted population count data v4.11 for 2015 from NASA SEDAC (sedac.ciesin.columbia.edu). Although we must still apply national mortality and population data from the World Bank in order to scale these results to 2014, we find that the gridded product reduces our estimate of ozone-attributable deaths to ~260 deaths in March and ~160 deaths in September across Southeast Asia. These numbers are somewhat lower than initially reported but still support our major conclusion that biomass burning coincides with widespread ozone exposure across Southeast Asia at levels that are dangerous to human health. We have updated the number of ozone-attributable deaths throughout the text as well as the description of the corresponding calculation at Line 372 on Page 12:

*"When we apply this factor to the gridded UN-adjusted population count product v4.11 for 2015 from the NASA Socioeconomic Data and Applications Center (sedac.ciesin.columbia.edu), scaled to 2014 using national mortality and population data from the World Bank (data.worldbank.org), we estimate that excessive ozone from biomass burning caused nearly 260 and 160 excess deaths across Southeast Asia in March and September, respectively."*

**Line 387: Is possible to add the GC simulated ozone concentrations as background in a) and b)? Hard to tell if GC capture the pattern of ozone when comparing Figures 10 and 11 which do not have the same map ratio.**

In Fig. 11, panels c and d show GEOS-Chem simulated ozone concentrations that are directly comparable against the ground observations shown in panels a and b. We recommend comparing between these panels rather than comparing against Fig. 10, which presents the same values but with a different color bar as well as a different map ratio, as mentioned by the referee.

**Figures:**
**Figure 2. '(b-e)' should be '(b-c)' for Spatial distribution of dry matter emissions?**

Yes, we have made this change and updated the captions for Fig. 2 and Fig. 4 to differentiate between the spatial distribution versus a map of emission types:

"**Figure 2.** *Monthly dry matter emissions (kg $m^{-2}$ $s^{-1}$) for Southeast Asia in 2014, as estimated by the GFED4.1s biomass burning inventory. (a) Timeseries of dry matter emissions. (b, c) Spatial distribution of dry matter emissions in March and September. (d, e) Map of the predominant vegetation types burned in March and September. Vegetation types shown in (a, d, e) include deforestation (DEFO), peat (PEAT), savanna (SAVA), agricultural waste (AGRI) and temperate forest (TEMF). There are no emissions from boreal forests (BORF) in this region.*"

"**Figure 4.** *Monthly mean emissions (kg $m^{-2}$ $s^{-1}$) of $NO_x$ and NMVOCs across Southeast Asia in March and September of 2014. (a, b, e, f) Spatial distribution of emissions. (c, d, g, h) Map of the predominant emission source types: anthropogenic, biogenic or pyrogenic.*"

**Figure 7. What are the resolution of GC and OMI? If they are gridded, why not show contours? The current plots look like there are lots of gaps.**

The resolution of the ozone columns in Fig. 7 is 0.25° × 0.3125°, consistent with the nested grid from GEOS-Chem. The model is filtered to show ozone columns only where they are observed by OMI and is otherwise replaced by white space. We prefer not to use contours here, to avoid suggesting that OMI data exists where it does not. However, we have updated the figure caption to clarify the resolution of these plots, the filtering procedure, and the meaning of the white space:

"**Figure 7.** *Spatial distribution of (a, d) GEOS-Chem model and (b, e) OMI monthly mean tropospheric ozone column (molec $cm^{-2}$) across Southeast Asia in March and September of 2014. Panels c and f show the difference between GEOS-Chem and OMI for March and September, respectively. Ozone columns are shown on the 0.25◦ x 0.3125◦ nested grid and are represented by white space where OMI observations are not available.*"